# PhpC^NF-Y transcription factor infiltrates heterochromatin to generate cryptic intron-containing transcripts crucial for small RNA production

Manjit Kumar Srivastav[1], H. Diego Folco[1], Patroula Nathanailidou [1],
Anupa T Anil[1], Drisya Vijayakumari[1], Shweta Jain[1], Jothy Dhakshnamoorthy[1],
Maura O'Neill[2], Thorkell Andresson[2], David Wheeler [1] & Shiv I. S. Grewal [1]✉

The assembly of repressive heterochromatin in eukaryotic genomes is crucial for silencing lineage-inappropriate genes and repetitive DNA elements. Paradoxically, transcription of repetitive elements within constitutive heterochromatin domains is required for RNA-based mechanisms, such as the RNAi pathway, to target heterochromatin assembly proteins. However, the mechanism by which heterochromatic repeats are transcribed has been unclear. Using fission yeast, we show that the conserved trimeric transcription factor (TF) PhpC^NF-Y complex can infiltrate constitutive heterochromatin via its histone-fold domains to transcribe repeat elements. PhpC^NF-Y collaborates with a Zn-finger containing TF to bind repeat promoter regions with *CCAAT* boxes. Mutating either the TFs or the *CCAAT* binding site disrupts the transcription of heterochromatic repeats. Although repeat elements are transcribed from both strands, PhpC^NF-Y-dependent transcripts originate from only one strand. These TF-driven transcripts contain multiple cryptic introns which are required for the generation of small interfering RNAs (siRNAs) via a mechanism involving the spliceosome and RNAi machinery. Our analyses show that siRNA production by this TF-mediated transcription pathway is critical for heterochromatin nucleation at target repeat loci. This study reveals a mechanism by which heterochromatic repeats are transcribed, initiating their own silencing by triggering a primary cascade that produces siRNAs necessary for heterochromatin nucleation.

Eukaryotic genomes are organized into "open" and "closed" chromatin domains referred to as euchromatin and heterochromatin[1,2]. Heterochromatin represses gene expression to ensure proper cellular differentiation and safeguards genome stability by silencing repetitive DNA elements[3,4]. The assembly of heterochromatin domains involves posttranslational modification of histones, which creates regions with characteristic hypoacetylation of histones and high levels of histone H3 lysine-9 methylation (H3K9me)[2,5]. Heterochromatin is nucleated at specific sites, known as nucleation sites, and then spread to surrounding sequences[2]. Evidence suggests that transcripts from target

---

[1]Laboratory of Biochemistry and Molecular Biology, National Cancer Institute, National Institutes of Health, Bethesda, MD, USA. [2]Cancer Research Technology Program, Frederick National Laboratory for Cancer Research, Frederick, MD, USA. ✉e-mail: grewals@mail.nih.gov

loci, including those derived from repetitive DNA elements, play an important role in guiding heterochromatin assembly proteins to nucleation sites[1,2,6]. However, despite the need for transcription within repressive heterochromatin domains, the mechanisms involved are not fully understood.

The fission yeast *Schizosaccharomyces pombe* is a valuable model system for studying heterochromatin assembly, as it shares conserved factors and pathways with higher eukaryotes. The Clr4 histone methyltransferase, like its mammalian counterparts SUV39H1 and SUV39H2, methylates histone H3K9[7,8], which in turn recruits HP1 chromodomain proteins[7,9–12]. In addition to methylating H3K9, Clr4[Suv39h] can recognize preexisting H3K9me3 through its chromodomain, and its ability to both "read" and "write" H3K9me is crucial for spreading and epigenetic inheritance of heterochromatin[2,13]. Spreading of H3K9me and its associated HP1 proteins allows heterochromatin-associated effectors, such as histone deacetylases (HDACs) involved in transcriptional gene silencing, to exert their influence across extended chromosomal domains[2].

Genome-wide mapping studies have discovered heterochromatin domains widely distributed across the *S. pombe* genome. While small heterochromatin islands are found at various regulated genes, the major targets of heterochromatin assembly are the pericentromeric repeat regions, sub-telomeres, and the silent mating-type (*mat*) region[14,15]. These constitutive heterochromatin domains are coated with H3K9me and HP1 proteins and each of them contains a specific class of repeat elements, referred to as *dg* and *dh* repeats, which serve as RNAi-dependent heterochromatin nucleation centers[16–18]. The silent *mat* region contains a single copy of the *dg*- and *dh*-like element, referred to as *cenH*, which serves as an RNAi-dependent heterochromatin nucleation center[16,19]. Heterochromatin targeted to *cenH* and a nearby site spreads across the ~20-kb domain surrounded by inverted repeat boundary elements (*IR-R* and *IR-L*) through the Clr4[Suv39h] read-write mechanism[13,16,20].

RNAi-mediated heterochromatin assembly necessitates the transcription of repeat elements by RNA polymerase II (RNAPII)[21,22]. During the S-phase, *dg/dh* repeats are transcribed bidirectionally[23,24], producing transcripts that are converted into siRNAs by the RNAi factors Argonaute (Ago1), Dicer (Dcr1), and RNA-dependent RNA polymerase (Rdp1)[14,25–28]. The siRNAs are incorporated into the RNA-induced transcriptional silencing (RITS) complex, comprising Ago1, Tas3, and Chp1[28], which is then guided by the siRNA to nascent repeat transcripts to recruit Clr4[Suv39h], thereby methylating H3K9[13,29,30].

A noteworthy aspect of RNAi-mediated heterochromatin assembly is a self-reinforcing loop: siRNAs generated by RNAi facilitate heterochromatin assembly, promoting the stable association of RNAi pathway proteins with repeat loci and ensuring efficient transcript processing[26,31]. Together, H3K9me and Swi6[HP1] play an important role in this process by engaging the RNA-dependent RNA polymerase complex (RDRC) to convert repeat transcripts into double-stranded RNAs for siRNA production by Dcr1[11,26,31–33]. However, substantial levels of siRNAs can still be produced in the absence of heterochromatin modifications[32,34]. This H3K9me- and Swi6[HP1]-independent processing of *dg/dh* transcripts by RNAi is considered essential for the initial heterochromatin nucleation. The precise mechanism by which RNAi machinery initiates siRNA accumulation independently of heterochromatin remains to be fully elucidated.

Here, we focused on identifying factors driving the transcription of repeat elements within repressive heterochromatin and discovered how the resulting transcripts are processed by the RNAi machinery to produce siRNAs. We identify a transcription factor (TF) complex, PhpC, analogous to the human NF-Y trimeric TF complex, that can infiltrate repressive heterochromatin via its histone-fold domain and promotes transcription of repeat elements. PhpC, which recognizes *CCAAT* box motifs, acts in a cooperative manner with another TF, named Moc3, that we show also localizes to *cenH* and *dh* repeats,

regardless of the heterochromatic state or cell cycle. Our work further reveals that TF-mediated transcription of heterochromatic repeats produces transcripts that contain inefficiently spliced introns, termed cryptic introns. Splicing machinery is required to engage RNAi machinery to these cryptic intron-containing transcripts to generate siRNAs and trigger heterochromatin nucleation. This study advances our understanding of the mechanisms that allow heterochromatic repeats to be transcribed to generate siRNAs for RNAi-mediated heterochromatin nucleation.

## Results

### TFs bind to repeat elements within heterochromatin domains

Although heterochromatin is generally inaccessible to transcriptional machinery, the transcription of heterochromatic repeat elements is essential for RNAi-mediated targeting of heterochromatin[21–24]. This raises a long-standing question: how are these elements transcribed?

To answer this question, we mapped the distribution of TFs, including those that show genetic interactions with the RNAi machinery and components of the Clr4[Suv39h] complex[35]. We focused on detecting binding to heterochromatic repeat elements, such as those at centromeres and the silent *mat* region, by performing chromatin immunoprecipitation-sequencing (ChIP-seq) analysis of TFs tagged with GFP. Atf1 and Pcr1, ATF-CREB family proteins previously known to bind specific sites at the silent *mat* region[36], were included as controls. As expected, our analyses detected TFs localizing to gene promoters, lncRNAs, and other ncRNAs (*tRNAs* and *rRNAs*) (Supplementary Fig. 1a and Supplementary Data 1). Notably, the histone fold domain-containing TF Php5[37,38] and a Zn finger type DNA binding protein Moc3[39] also showed significant enrichment at the *cenH* element within the silent *mat* region and the *dh* centromeric repeat elements adjacent to siRNA hotspots (Fig. 1a, b and Supplementary Fig. 1b). The binding peak of Php5 overlapped with that of Moc3 but not with those of Atf1 and Pcr1 that localize to the previously described *CAS* (Clr3-attracting sequence) element (Fig. 1c and Supplementary Fig. 1c)[40].

Since *cenH* shares approximately 96% homology with centromeric *dg* and *dh* repeats[19], ChIP-seq analysis cannot conclusively determine if TFs localize to both or only one of the locations. In fact, the observed *cenH* peak contains a mix of reads that map uniquely to *cenH* in addition to some reads that map equally well to pericentromeric *dg/dh*. Therefore, to further validate our ChIP-seq results, we performed site-specific qPCR using primers specific to each of the heterochromatic locations adjacent to the siRNA hotspots. By using this method, we confirmed that Php5 and Moc3 were indeed enriched at both *cenH* and *dh* loci (Fig. 1d and Supplementary Fig. 1b). These results indicate that TFs can access repeat elements located within repressive heterochromatin domains.

### Heterochromatic TFs exhibit partial colocalization across the genome

The co-localization of Php5 and Moc3 at *cenH* and *dh* heterochromatic repeats suggested a broader collaboration between these factors. To further explore this, we performed a comparative analysis of the genome-wide distribution patterns of these TFs alongside others. A hierarchical clustering based on the signal strength of the TFs at 2,273 binding sites identified six distinct clusters (Supplementary Fig. 1d). Atf1 and Pcr1, known for their role in stress response[41], were grouped within one cluster (Supplementary Fig. 1d). Despite their colocalization at heterochromatin repeat elements, Php5 and Moc3 were found in separate clusters. Only ~24% (180/762) of Moc3-occupied sites, including several ncRNAs, also contained Php5, while approximately 46% (180/389) of Php5-occupied peaks also had Moc3 (Supplementary Fig. 1e, f and Supplementary Data 2). Gene Ontology (GO) analysis revealed both Moc3 and Php5 bind to the promoters of target genes that are linked to various pathways involved in energy production, such as NADH metabolic processes, ATP generation, and glycolytic

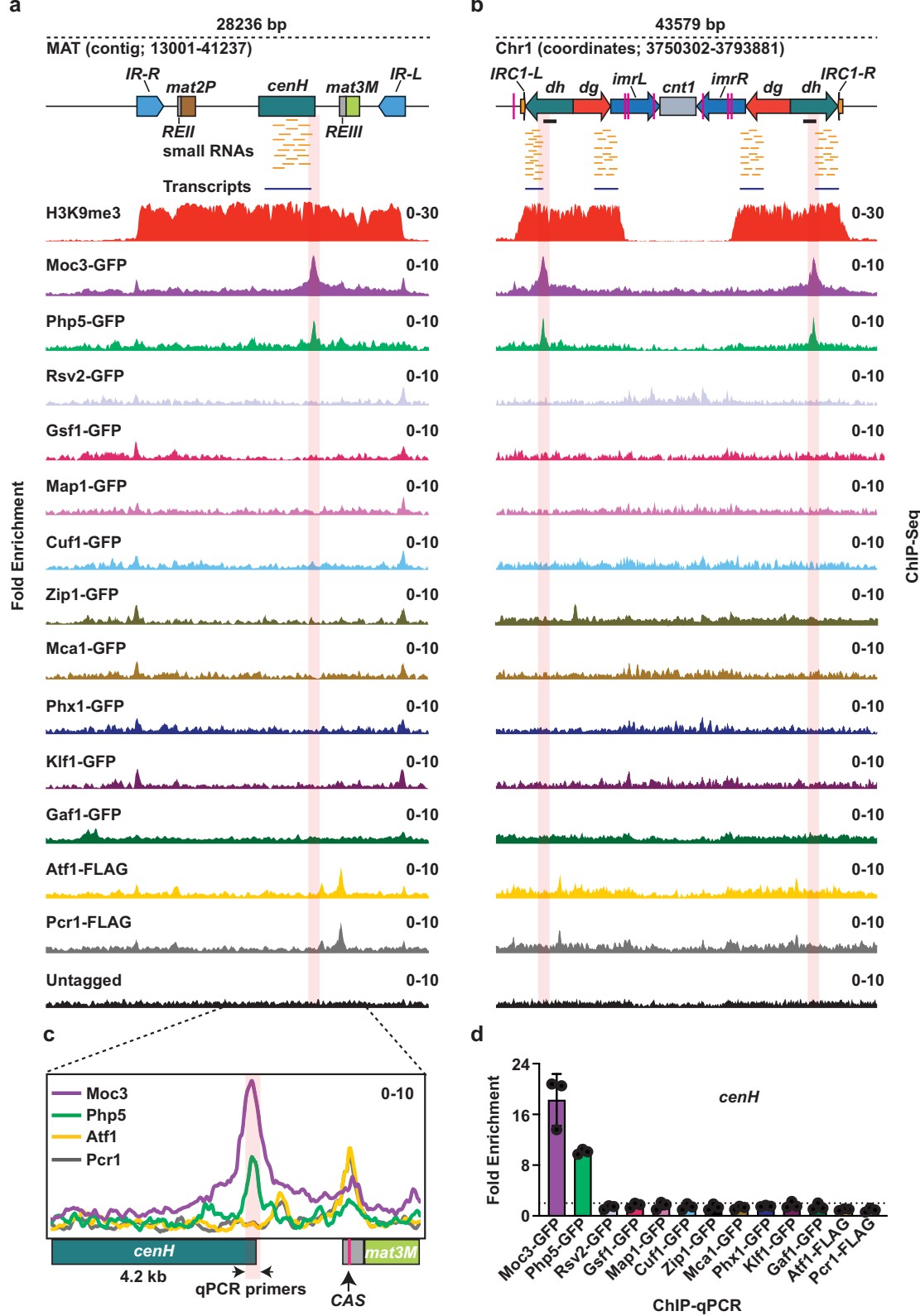

**Fig. 1 | The transcription factors Php5 and Moc3 bind to heterochromatic repeats. a**, **b** ChIP-seq analysis of TF distribution at the silent *mat* locus (**a**) and centromere 1 (**b**) in a wild-type background strain. The blue and small orange lines represent transcripts and small RNAs, respectively. Annotations: *cenH* refers to homology to centromeric *dg/dh*; IR signifies an Inverted Repeat. **c** The enlarged section from (**a**) shows the distribution of the specified TFs. *CAS* denotes Clr3-attracting sequence. **d** ChIP-qPCR analysis of the distribution of the indicated GFP-tagged transcription factors at *cenH* in wild-type cells. The data is expressed as the relative fold enrichment compared to the *leu1* control locus and is normalized to the untagged strain. Significant enrichments are defined as those ≥ 2-fold (dotted line). Data from 3 independent biological experiments are presented as the mean ± SD. The positions of qPCR primers for *dh* and *cenH* are indicated by the black lines and arrows in (**b**) and (**c**), respectively. Source data are provided as a Source Data file.

pathways (Supplementary Fig. 1g and Supplementary Data 2). Overall, these findings suggest that although Php5 and Moc3 colocalize at sites within constitutive heterochromatin domains, their colocalization in other genomic regions is only partial. This indicates potential shared as well as distinct functions of these factors.

## Php5 forms a trimeric complex that localizes to heterochromatic repeats

Unlike Moc3, which is present only in *Schizosaccharomycetes*, Php5 is highly conserved from budding yeast (Hap5) and pathogenic yeast *Candida albicans* (Hap5) to humans (NF-YC). In other organisms, Php5 orthologs are known to associate with two other proteins, *Hs*NF-YA/*Sc*Hap2/*Ca*Hap2 and *Hs*NF-YB/*Sc*Hap3/*Ca*Hap3, to form a trimeric complex[38,42–45]. Genetic evidence indicates that a trimeric complex likely also exists in *S. pombe*, but this has not been biochemically confirmed. We therefore focused on identifying factors that are associated with Php5. For this, we performed immuno-affinity purification of GFP-tagged Php5 (Fig. 2a) and analyzed the purified fraction by mass spectroscopy. In addition to Php5, these analyses identified NF-YA and NF-YB orthologs, Php2 and Php3, respectively (Fig. 2b). Therefore, Php5 associates with Php3[NF-YB] and Php2[NF-YA] to form the Php complex (PhpC), as observed in other yeasts and humans. Interestingly, our analyses also identified Moc3, as well as Atf1 and Pcr1, in the Php5 purified fraction (Fig. 2b). Moreover, several other proteins, including the components of RSC and Ino80 remodeler complexes, also co-purified with Php5 (Supplementary Fig. 2a).

We next investigated whether Php2 and Php3 bind at the heterochromatic regions where Php5 is found. ChIP-seq analyses revealed that both proteins co-localized with Php5 at the *cenH* and *dh* repeats adjacent to siRNA hotspots, and this result was confirmed by ChIP-qPCR (Fig. 2c–f). Notably, the enrichment level of Php2 at both locations was lower than that of the Php3 and Php5 subunits. The reason for this enrichment difference is unclear, as live-cell imaging indicated similar expression levels for all PhpC subunits (Supplementary Fig. 2b).

The trimeric NF-Y complex and its counterparts recognize a pentanucleotide motif called the *CCAAT* box[42–44]. This led us to ask if the PhpC subunits are also associated with chromosomal regions containing this motif. To address this, we counted the *CCAAT* motifs on both strands within -/+ 500 bp of the peak center, and as previously proposed, motifs separated by ≤25 bp were counted as single ones[46]. A cumulative total of 920 genomic binding peaks were found for Php2, Php3, and Php5. Notably, approximately 73% (673/920) of Php2, Php3, and Php5 peaks correspond to genomic regions containing more than one *CCAAT* box, including those at the *cenH* and *dh* loci (Fig. 2g, h and Supplementary Fig. 2c). The levels of Php proteins were relatively higher at loci where the number of *CCAAT* boxes was greater (Fig. 2i, j and Supplementary Data 3). In particular, *CCAAT* boxes were especially enriched at loci simultaneously occupied by all three PhpC subunits (Fig. 2g). For example, PhpC subunits localize to the *CCAAT* box-containing locus *sme2*, a lncRNA essential for sexual development[47], and to the promoter regions of known target genes such as *cyc1*+ and *pcl1*+ (Supplementary Fig. 2d, e)[38,48]. Taken together, these results indicate that Php2[NF-YA], Php3[NF-YB], and Php5[NF-YC] form a trimeric complex that localizes to numerous loci containing the *CCAAT* box motif, including the *cenH* and *dh* heterochromatic repeat elements.

## TF binding to heterochromatic repeats is interdependent

We next asked if PhpC and Moc3 independently localize to heterochromatic regions or if their localization is mutually dependent. We performed ChIP-seq and ChIP-qPCR analyses of PhpC subunits and Moc3 in strains where one of the other factors was deleted. Consistent with the fact that Php2/3/5 forms a complex, loss of Php3 localization at *cenH* and *dh* elements was observed in the absence of *php2* or *php5* (Fig. 3a, b, and Supplementary Fig. 3a). Likewise, the localization of

Php2 and Php5 at these sites decreased upon the deletion of any other PhpC subunit (Supplementary Fig. 3b, c). Interestingly, we also observed that Moc3 occupancy at both *cenH* and *dh* heterochromatic regions was contingent upon PhpC (Fig. 3a, c and Supplementary Fig. 3d). Moreover, the localization of PhpC subunits was compromised in *moc3Δ* cells (Fig. 3a, b and Supplementary Fig. 3b, c). The nuclear localization of PhpC and Moc3 persisted upon loss of any of the other factors (Supplementary Fig. 3e), consistent with the fact that these proteins still independently localize to a significant fraction of euchromatic locations (Supplementary Fig. 3f–h and Supplementary Data 4). These findings suggest that the TFs act in a cooperative manner to localize to heterochromatic repeat elements.

## The *CCAAT* box is required for PhpC heterochromatin localization

The regions within *cenH* and *dh* elements where TFs bind contain two *CCAAT* boxes that are separated by approximately ~60 bp. To investigate the significance of these elements for the heterochromatin association of TFs, we modified both *CCAAT* boxes in *cenH* to *ATGAC* (Supplementary Fig. 3i, hereafter referred to as *CCAAT*[mut]). ChIP-seq and ChIP-qPCR analyses revealed a drastic decrease in the binding of PhpC subunits at *cenH* in *CCAAT*[mut] cells (Fig. 3d, e and Supplementary Fig. 3j), whereas the localization of PhpC at the centromeric *dh* elements (Fig. 3d, e and Supplementary Fig. 3j) and euchromatic locations (Supplementary Fig. 3k, and Supplementary Data 4), where *CCAAT* boxes remained unchanged, were unaffected. Consistent with our results showing that PhpC is required for Moc3 localization at heterochromatic repeats, *CCAAT*[mut] cells were defective in Moc3 binding at *cenH* (Fig. 3d, f). These results emphasize the importance of *CCAAT* boxes in guiding TFs to specific sequences within a heterochromatin domain at the silent *mat* region.

## Heterochromatic localization of TFs is independent of the cell cycle

Previous studies have shown that transcription from heterochromatic repeats occurs predominantly during S-phase[23,24], suggesting that transcription factors may access repeat loci specifically during this cell cycle phase. However, our ChIP analyses were conducted in asynchronously growing cells, making it difficult to determine if only the cells in the S-phase show TF enrichment at heterochromatic repeats. The human counterpart of PhpC, NF-Y, remains bound to chromatin during mitosis[49]. To address whether PhpC and Moc3 remain similarly bound to chromatin, we performed ChIP-seq analyses in cells that carry a cold-sensitive mutation in the β-tubulin gene (*nda3-KM311*), which causes mitotic prophase arrest at low-temperature[50]. Php3 and Moc3 continued to associate with heterochromatic repeat elements even when cells were shifted to low temperature (Fig. 4a–d and Supplementary Fig. 4a, b). Furthermore, the genome-wide distribution of Php3 and Moc3 remained largely unchanged in mitotic-arrested cells (Supplementary Data 4), indicating that PhpC and Moc3 maintain their association with heterochromatin outside of S-phase.

We further investigated the impact of heterochromatin on TF occupancy across heterochromatic regions. PhpC and Moc3 peaks at *cenH* and *dh* elements were mostly unchanged in cells lacking Clr4[Suv39h] (Fig. 4a–d and Supplementary Fig. 4a, b). Interestingly, the absence of Clr4[Suv39h] resulted in the emergence of additional minor peaks of PhpC, and its binding extended across *dh* and part of the *dg* repeat at centromeres (Fig. 4b and Supplementary Fig. 4c). However, global Php3 and Moc3 binding at gene promoters, lncRNAs, and other ncRNAs remained unchanged in *clr4Δ* cells (Supplementary Data 4). Therefore, heterochromatin disruption allows the binding of transcription factors to other sites within heterochromatic regions.

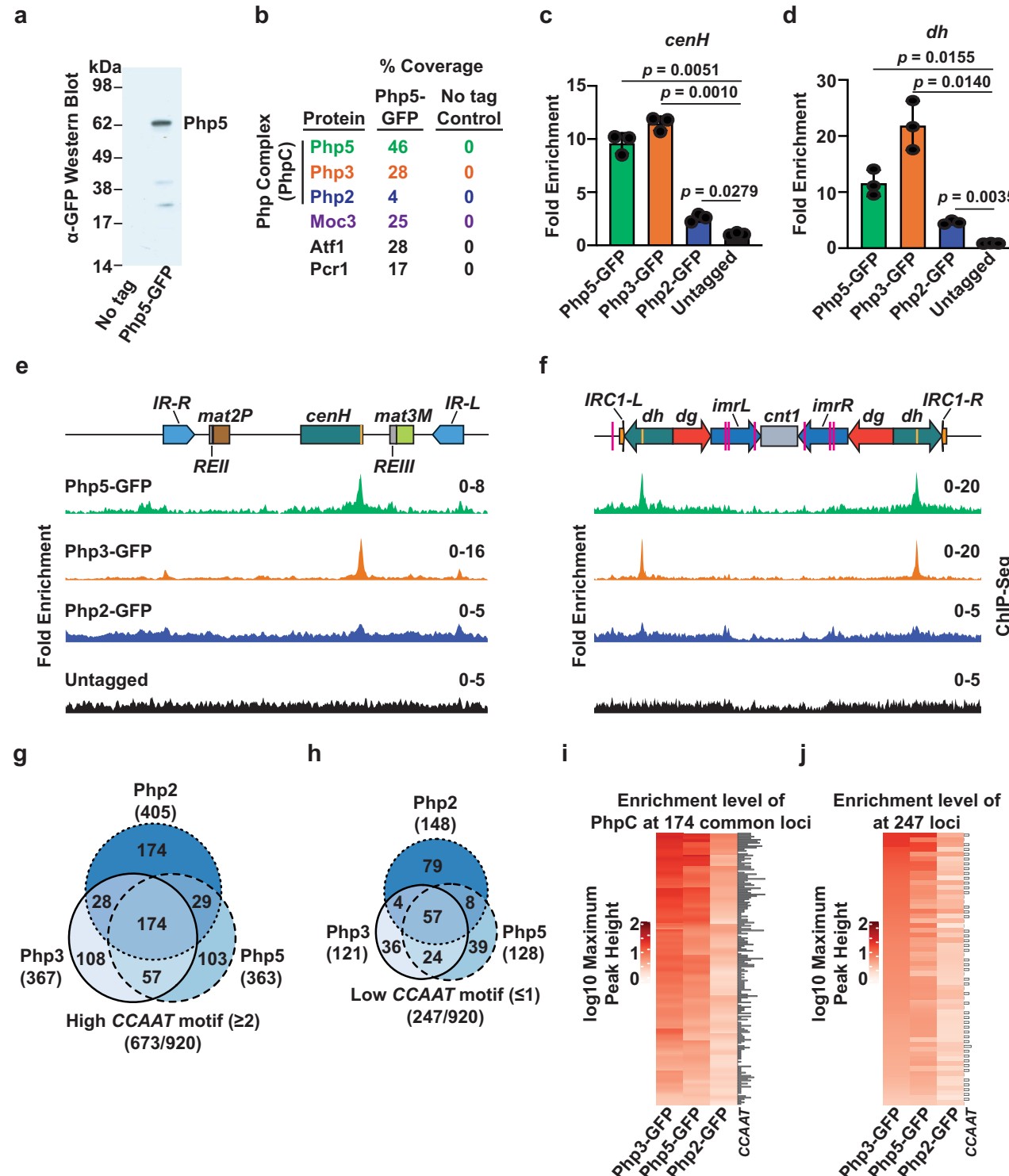

**Fig. 2 | Php5 forms a trimeric complex that localizes to heterochromatic repeats. a** Php5 immunopurified fractions were prepared from untagged and Php5-GFP expressing cells and were probed with anti-GFP antibodies. **b** Immunopurified fractions from untagged and Php5-GFP expressing cells were subjected to mass spectrometry analysis. The percentage of total peptide coverage for the indicated immunopurified proteins is displayed. Results from one biological replicate are shown; see also Supplementary Fig. 2a. **c**, **d** ChIP-qPCR analysis of the indicated GFP-tagged TFs was performed at *cenH* (**c**) and *dh* (**d**) in wild-type cells. Data from 3 independent biological experiments are presented as the mean ± SD of the relative

fold enrichment compared to the *leu1* control locus. The *p*-values were calculated using a two-tailed paired t-test. **e**, **f** ChIP-seq analysis to determine the localization of the indicated GFP-tagged TFs at the silent *mat* locus (**e**) and centromere 1 (**f**) in wild-type cells. Note that the Php5-GFP ChIP-seq data is the same as in Fig. 1a, b, as the experiments were performed simultaneously. **g**, **h** Euler diagrams represent the number of loci bound by Php5, Php3, and Php2 and contain a high (**g**) or low (**h**) abundance of *CCAAT* boxes. **i**, **j** Heat maps displaying ChIP-seq enrichments of PhpC subunits at loci containing high (**i**) or low (**j**) abundance of *CCAAT* boxes. Source Data are provided as a Source data file.

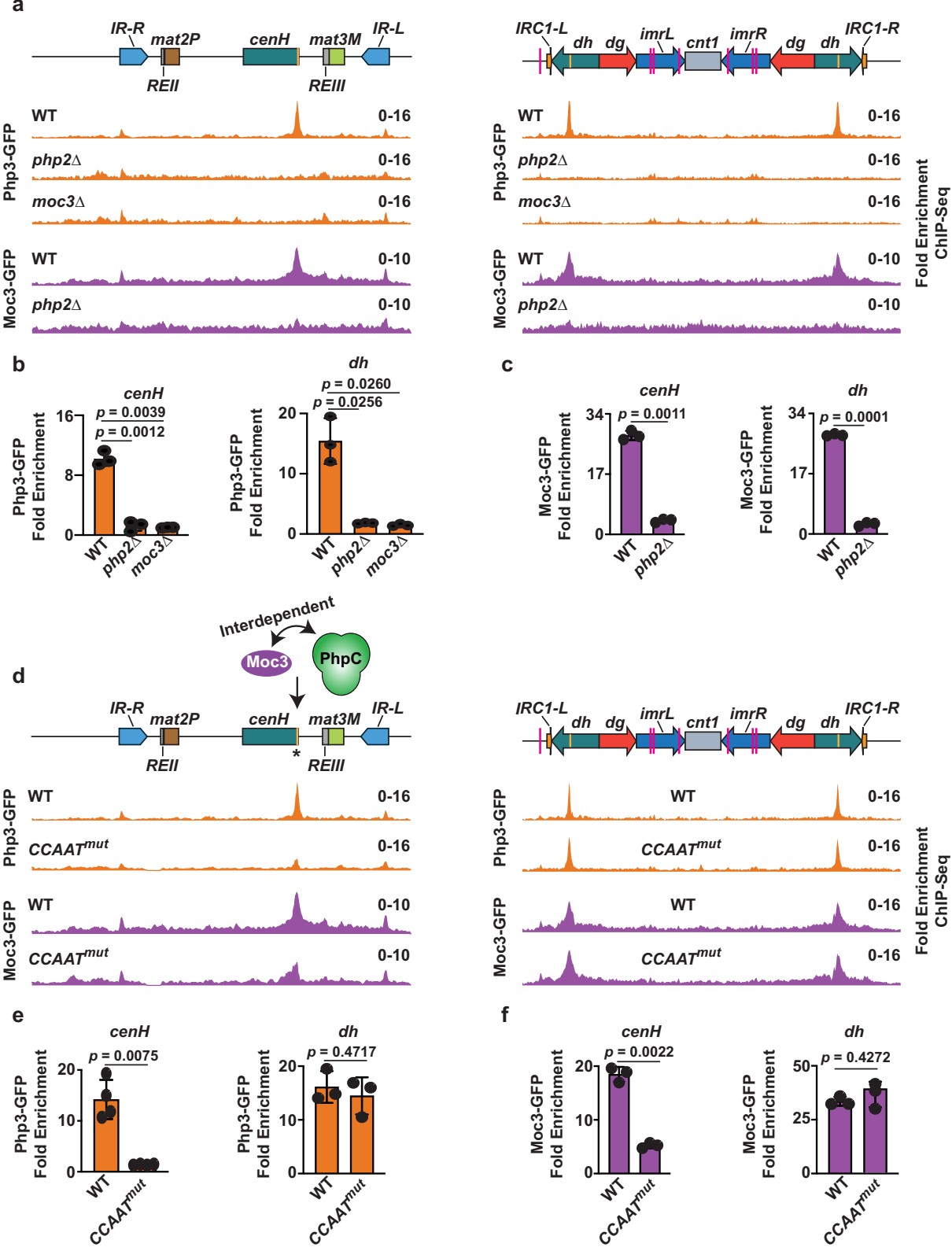

**Fig. 3 | The binding of PhpC and Moc3 to heterochromatin is interdependent and requires *CCAAT* boxes.** **a** ChIP-seq analysis of GFP-tagged Php3 and Moc3 at the silent *mat* locus and centromere 1 in the indicated strains. **b, c** Fold enrichments of GFP-tagged Php3 and Moc3 were determined by ChIP-qPCR analysis at *cenH* and *dh*. **d** ChIP-seq analysis of GFP-tagged Php3 and Moc3 expressed in wild-type or *CCAAT^mut* strains. *CCAAT* boxes are indicated by vertical yellow lines in the schematics above, and the asterisk represents the location of two mutated *CCAAT*

boxes. **e, f** Fold enrichments of GFP-tagged Php3 and Moc3 were determined by ChIP-qPCR analysis at *cenH* and *dh*. Data from 3 independent biological experiments are presented as the mean ± SD of the relative fold enrichment compared to the *leu1* control locus. The *p*-values were calculated using a two-tailed paired t-test (**b, c, e, f**). WT Moc3-GFP and Php3-GFP ChIP-Seq data are replotted from Figs. 1a, b and 2e, f (**a, d**). Source data are provided as a Source Data file.

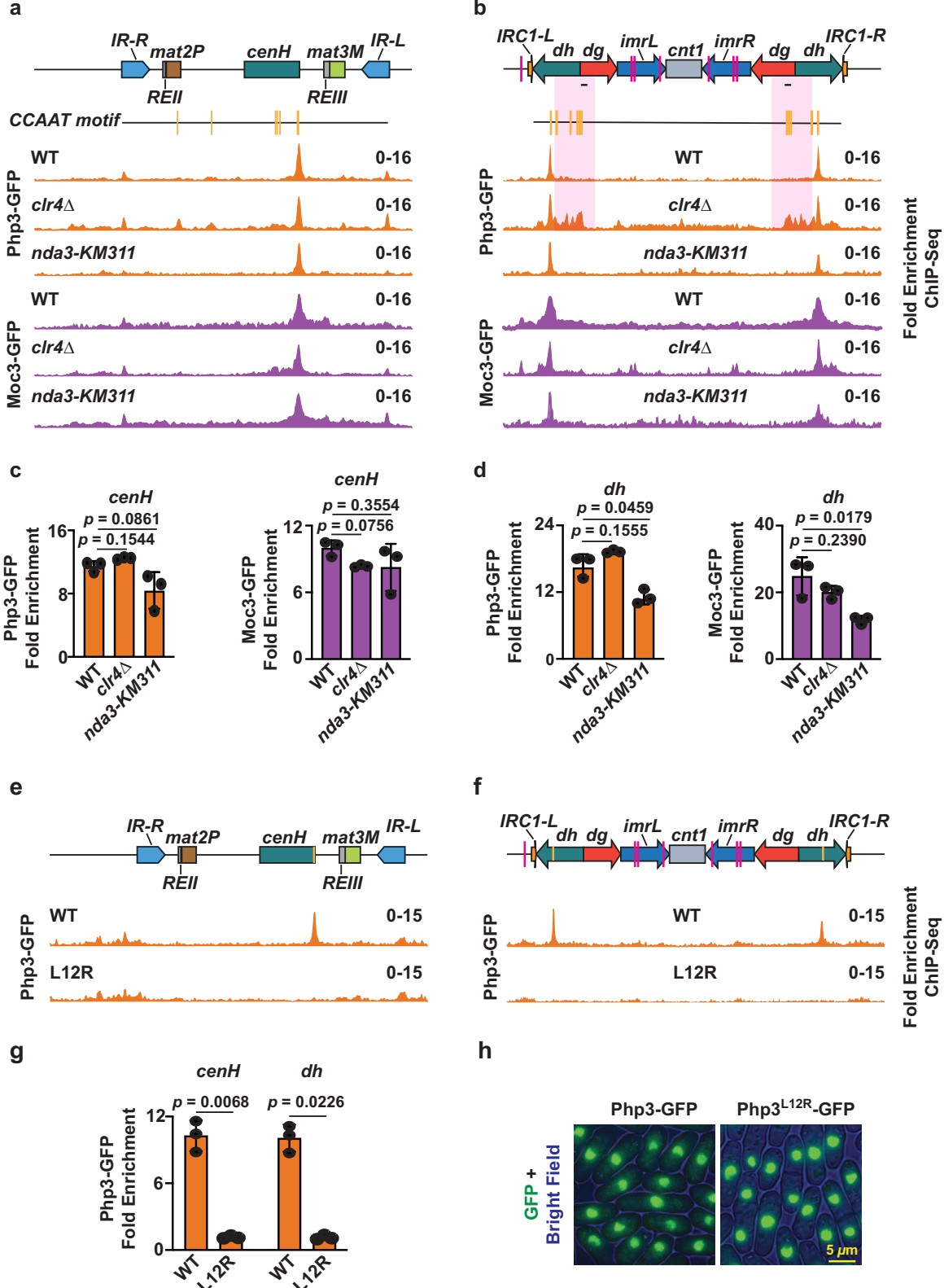

**Fig. 4 | The histone fold domain of PhpC is pivotal for heterochromatin localization. a, b** ChIP-seq analysis of Php3 and Moc3 localization at the silent *mat* locus (**a**) and centromere 1 (**b**) in wild type, *clr4Δ*, and *nda3-KM311* strains. WT Moc3-GFP and Php3-GFP ChIP-seq data are replotted from Figs. 1a, b and 2e, f. Vertical yellow lines designate *CCAAT* boxes. Note that the highlighted *CCAAT*-enriched region is accessible to Php3 in *clr4Δ* cells. The small black line denotes the *dg*-annealing primer used in Supplementary Fig. 4c. **c, d** ChIP-qPCR analysis of GFP-tagged Php3 or Moc3 was performed at *cenH* (**c**) and *dh* (**d**) in the indicated strains. **e, f** ChIP-seq

analysis of Php3 localization at the silent *mat* locus (**e**) and centromere 1 (**f**) in wild type and *php3*$^{L12R}$ strains. **g** ChIP-qPCR analysis of GFP-tagged Php3 was performed at *cenH* and *dh* in indicated strains. **h** Representative live-cell images of strains expressing GFP-tagged Php3 or Php3$^{L12R}$ from two independent experiments with similar results. Data from 3 independent biological experiments are presented as the mean ± SD. The *p*-values were calculated using a two-tailed paired t-test (**c**, **d**, **g**). Source data are provided as a Source Data file.

## TF can localize to heterochromatin without HATs and chromatin remodelers

Histone acetyltransferases (HATs) and chromatin remodelers play a key role in regulating chromatin accessibility and facilitating the binding of TFs to target gene loci[51]. To determine if HATs are required for TF localization at heterochromatic loci, we investigated the distribution of Php3 and Moc3 in cells devoid of both Gcn5, a subunit of the SAGA complex, and Mst2, a MYST acetyltransferase. Interestingly, we found that in those cells, both TFs can still localize to *cenH* and *dh* heterochromatic repeat elements, albeit with a slight reduction (Supplementary Fig. 4d, e). This result suggests that HAT activity has a minimal impact on TF localization to heterochromatin regions.

Since we identified components of the RSC and Ino80 chromatin remodelers in purified Php5 fractions, we also studied the possible roles of remodelers in TF heterochromatic binding (Supplementary Fig. 2a). Specifically, we assessed TF localization in cells lacking the RSC subunit Rsc1, the SWI/SNF ATP-dependent chromatin remodeler Snf22 and/or bearing a mutation in the ATPase domain of Ino80 (i.e., *ino80^{K873A}*). In all cases, we observed that Php3 and Moc3 binding to *cenH* and *dh* regions were only marginally reduced but nevertheless not abolished (Supplementary Fig. 4f–i). These findings suggest that PhpC and Moc3 can access heterochromatic regions independently of the HATs and chromatin remodeling factors examined.

## Histone-fold domains in PhpC subunits are vital for accessing heterochromatin

The PhpC subunits Php3 and Php5 contain H2B- and H2A-like histone-fold domains (HFDs), respectively (Supplementary Fig. 5a). We constructed strains bearing mutations of conserved residues of Php3 (L12R) and Php5 (A112D, R113D) HFDs to investigate the role of these domains in TF localization to heterochromatic loci. Remarkably, we observed that in both mutants, the binding of PhpC subunits Php3 and Php5 at *cenH* and *dh* heterochromatic repeats was drastically reduced (Fig. 4e–g, Supplementary Fig. 5b), while the nuclear localization of both TFs was preserved (Fig. 4h, Supplementary Fig. 5c). Indeed, site-specific ChIP-qPCR analysis revealed a ~90% reduction in Php3 and a ~80% reduction in Php5 in HFD mutants compared to WT (Fig. 4g, Supplementary Fig. 5b). On the other hand, the global binding of Php3 at gene promoters, lncRNAs, and other ncRNAs, was reduced but not abolished in HFD mutant cells (Supplementary Fig. 5d, e, Supplementary Data 4). Therefore, HFDs of PhpC subunits are necessary to facilitate TF accessibility to heterochromatic regions.

## TFs promote RNAPII transcription of *cenH* heterochromatic repeats

As mentioned above, RNAPII transcription of heterochromatic repeat elements is crucial for heterochromatin nucleation. Since PhpC and Moc3 localize to heterochromatic repeats, we investigated if these TFs promote transcription at these sites. To avoid complications associated with the analysis of multicopy repeat elements, we exploited previously described sequence polymorphisms[19] to focus primarily on transcripts derived from the *cenH* element. In wild-type (WT) cells, *cenH* transcripts are challenging to detect. To overcome this, we utilized cells lacking Clr4^{Suv39h}, wherein transcription is enhanced and transcripts are easier to detect. As expected, we observed increased *cenH* transcript levels in *clr4Δ*[26] (Fig. 5a, b and Supplementary Fig. 6a, b). Interestingly, the lack of PhpC significantly decreased *cenH* transcript levels in *clr4Δ* (Fig. 5a, b). Similar changes were observed in *moc3Δ* (Fig. 5a and Supplementary Fig. 6b)[52]. Moreover, introducing *CCAAT^{mut}* into *clr4Δ* cells led to a notable reduction in *cenH* transcript levels (Fig. 5a and Supplementary Fig. 6a). These results were confirmed by RNA-seq analyses. Notably, the loss of PhpC and Moc3, as well as the *CCAAT^{mut}* mutation in Clr4-deficient cells, caused a substantial reduction in transcripts originating from the bottom strand of the *cenH* element, while the levels of transcripts from the top strand

remained largely unchanged (Fig. 5c, d). These findings indicate that TFs play a significant role in driving transcription of the bottom strand of the *cenH* element, suggesting that their binding defines an active promoter responsible for transcribing downstream *cenH* sequences.

We then investigated the role of TFs in RNAPII occupancy at *cenH*. ChIP-seq analysis of the Rpb1 subunit of RNAPII was performed, and the results were confirmed by site-specific ChIP-qPCR. RNAPII occupancy increased at *cenH* in *clr4Δ* (Fig. 5e, f and Supplementary Fig. 6c). Removal of PhpC or Moc3, or the introduction of *CCAAT^{mut}* mutation in *clr4Δ* cells, resulted in a considerable reduction in RNAPII levels at *cenH* (Fig. 5e, f). Similar results were obtained using an antibody against RNAPII CTD Ser-2P (Supplementary Fig. 6c). As a control, we also examined the impact of *php2Δ* at a known Php target and confirmed reduced transcription and loss of RNAPII occupancy independent of Clr4^{Suv39h} (Supplementary Fig. 6d, e). Overall, these findings demonstrate the importance of TFs in promoting *cenH* transcription by RNAPII.

## TF-mediated *cenH* transcription supports Swi6^{HP1}-independent siRNA production

Since transcription of heterochromatic repeats by RNAPII is necessary for the generation of siRNAs by the RNAi machinery, we investigated whether TF binding to *cenH* is also crucial for siRNA production. As described above, two distinct mechanisms engage RNAi machinery to process repeat transcripts. In addition to H3K9me and Swi6^{HP1} recruiting RITS and RDRC to facilitate the conversion of repeat transcripts into siRNAs by Dcr1[26,31–33], RNAi machinery can also be targeted to repeat-derived RNAs via a Swi6^{HP1}-independent mechanism[32].

To determine the impact of *php3Δ*, *moc3Δ*, or *CCAAT^{mut}* on siRNA production from *cenH*, we conducted small RNA sequencing analysis in the presence or absence of Swi6^{HP1}. To rule out the contribution of centromeric siRNA due to the homology between *cenH* and *dg/dh* repeats, we analyzed only the reads that uniquely map to *cenH*. Single mutant cells carrying *php3Δ*, *moc3Δ*, or *CCAAT^{mut}* showed little or no change in *cenH* siRNAs compared to WT cells (Fig. 6a). On the other hand, loss of Swi6^{HP1}, which causes an increase in heterochromatic repeat transcription (Supplementary Fig. 7a), showed a corresponding increase in *cenH* siRNAs. However, combining *swi6Δ* with either *php3Δ*, *moc3Δ*, or *CCAAT^{mut}* led to a severe reduction or loss of siRNAs mapping to *cenH* (Fig. 6a). Together, these results suggest that while Swi6 can mediate siRNA generation from the top strand, TF-driven transcription of the *cenH* bottom-strand is required to produce Swi6^{HP1}-independent siRNAs.

## Processing of TF-generated transcripts into siRNAs requires the spliceosome and cryptic intron splice sites

The observation that PhpC-mediated transcription of the bottom *cenH* strand is required for Swi6^{HP1}-independent production of siRNAs led us to investigate the unique features of this transcript strand that may help recruit the RNAi machinery. Closer examination revealed that a region of *cenH* sharing homology to *dh* repeats contains multiple cryptic introns, which have been implicated in RNAi-mediated heterochromatin formation[53–55]. RNAi machinery interacts with the splicing factors that are necessary for proper heterochromatin formation[56]. Notably, these cryptic introns are predominantly found in the bottom transcript strand (Supplementary Fig. 7b).

We wondered whether the presence of multiple cryptic introns in these TF-driven transcripts engages splicing factors implicated in siRNA production by the RNAi machinery. To investigate this, we tested if splicing machinery is required for Swi6^{HP1}-independent production of siRNAs, similar to the TFs involved in bottom-strand transcription. A temperature-sensitive mutation in the conserved spliceosome component Cwf10^{EFTUD2} (*cwf10-1*)[56] alone caused only a modest defect in the production of siRNAs that map to *cenH* (Supplementary Fig. 7c). On the other hand, when *cwf10-1* was combined

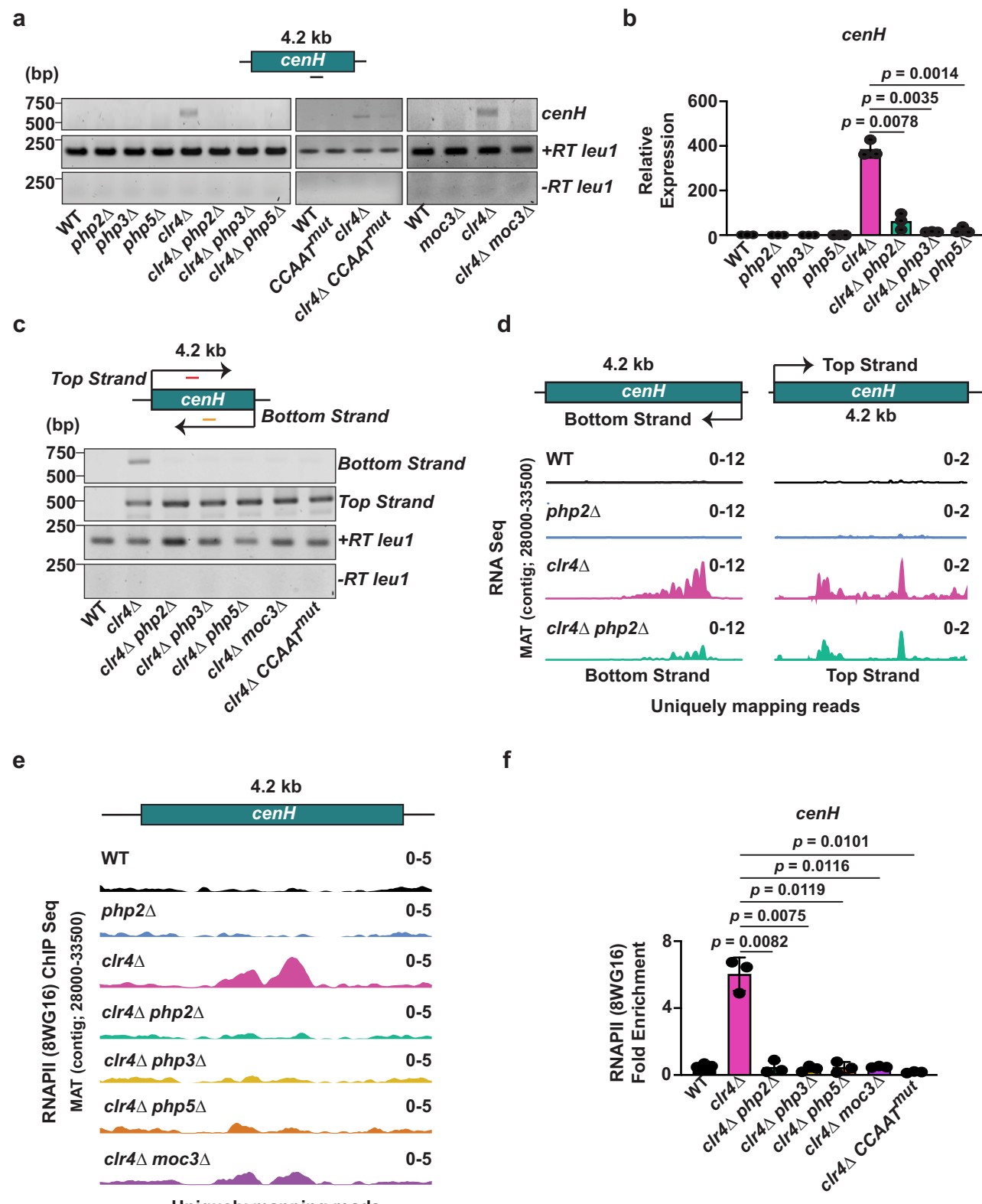

**Fig. 5 | Cooperative function of PhpC and Moc3 promotes *cenH* transcription.**
**a** RT-PCR was performed to analyze *cenH* transcripts in the indicated strains, with *leu1* serving as the control for +RT and -RT reactions. The primer binding site at *cenH* is indicated by the black line. **b** RT-qPCR analysis of *cenH* transcripts in the indicated strains. Relative expression was compared to the *leu1* control locus. **c** RT-PCR was performed to analyze *cenH* top and bottom strand transcripts in the indicated strains, with *leu1* used as the control for +RT and -RT reactions. The top and bottom strand-specific primer binding site at *cenH* is indicated by the red and yellow lines, respectively. Representative results from two independent experiments are shown. **d** RNA-seq expression profile of the *cenH* top and bottom strand in the indicated strains. Reads were uniquely mapped to *cenH*. **e, f** RNA polymerase II (8WG16) occupancy at *cenH* was determined by ChIP-seq (**e**) and ChIP-qPCR (**f**) analyses in the indicated strains. Reads were uniquely mapped to *cenH*. For **f**, ChIP data is presented as relative fold enrichment compared to the *tRNA* control locus. Data from 3 independent biological experiments are presented as the mean ± SD. The *p*-values were calculated using a two-tailed paired t-test (**b**, **f**). Source data are provided as a Source Data file.

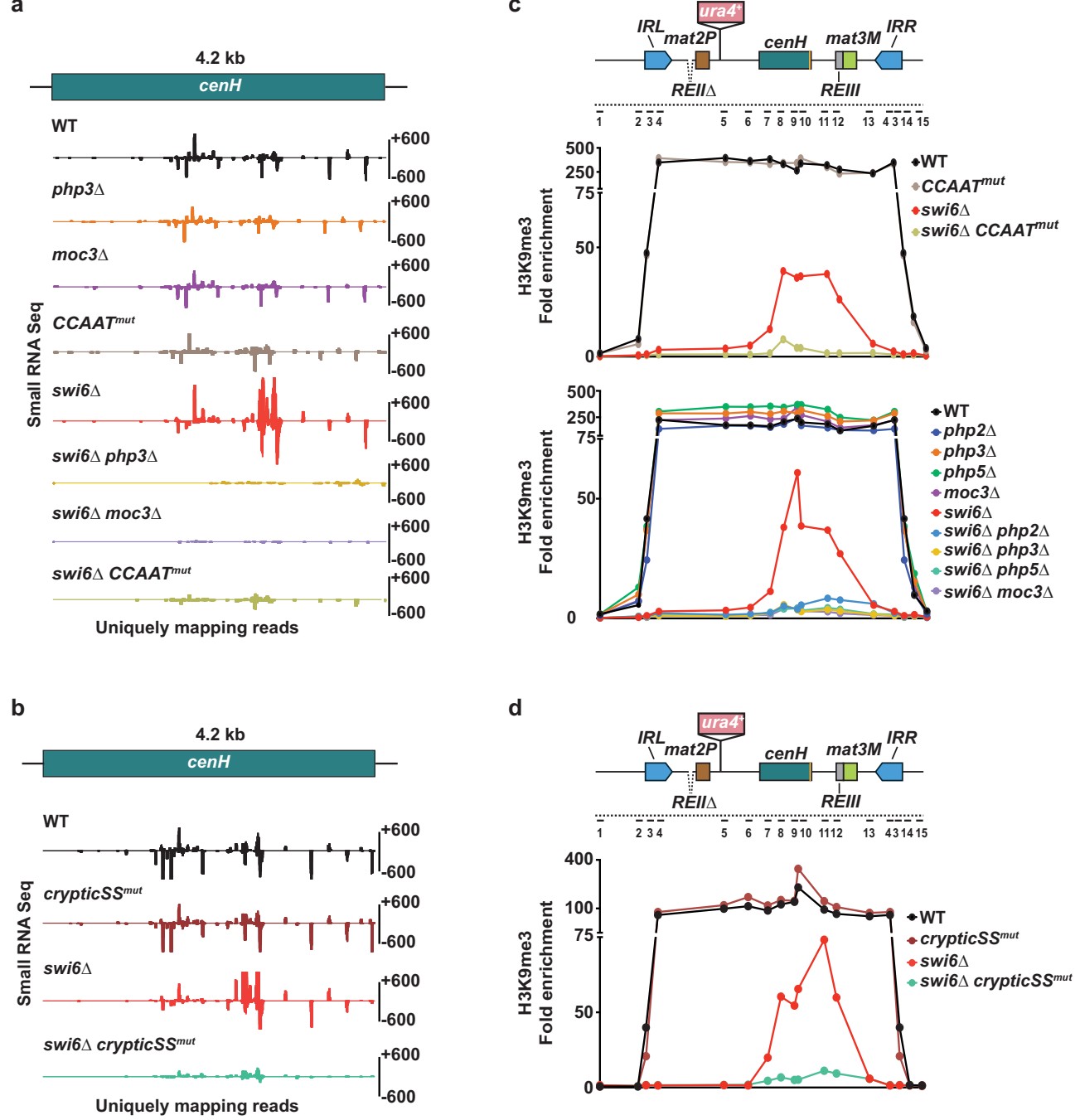

**Fig. 6 | PhpC- and Moc3-mediated *cenH* transcription is crucial for producing Swi6HP1-independent siRNAs. a**, **b** siRNA-seq profiles at the *cenH* region in the indicated strains. Note that the reads were uniquely mapped to *cenH*. **c**, **d** ChIP-qPCR analysis of H3K9me3 enrichment at the silent *mat* locus in the indicated strains. Data is presented as the mean of 2 independent biological experiments. The numbered lines in the schematic denote the location of the primers. Source data are provided as a Source data file.

with *swi6Δ*, the double mutant cells showed a dramatic reduction in *cenH* siRNAs (Supplementary Fig. 7c).

We next mutated the splice sites of all five cryptic introns found in the *cenH* bottom strand (Supplementary Fig. 7b, hereafter referred to as *crypticSSmut*). Whereas mutating the cryptic intron splice sites alone did not cause major changes in the levels of siRNAs mapping to *cenH*, combining *crypticSSmut* with *swi6Δ* resulted in a severe reduction of *cenH* siRNAs (Fig. 6b). Together, these results suggest that cryptic introns in the TF-driven *cenH* transcripts engage RNAi machinery via splicing machinery to generate Swi6HP1-independent siRNAs.

## H3K9me targeting to *cenH* requires TF-driven transcription

The siRNAs are believed to provide specificity for RNAi-mediated targeting of Clr4Suv39h to methylate H3K9[2]. At the silent *mat* region, heterochromatin targeted to *cenH* by RNAi spreads in *cis* across the entire silenced domain via a mechanism involving Swi6HP1 and the read-write activity of Clr4Suv39h [13,16]. In cells defective in heterochromatin spreading, such as the *swi6* mutant, H3K9me is mainly restricted to the *cenH* region and fails to spread to surrounding sequences[16].

We explored the contribution of the TF-driven transcription of *cenH* in heterochromatin nucleation by mapping H3K9me3

throughout the silent *mat* region in the *swi6Δ* background. As expected, the loss of Swi6[HP1] effectively limited H3K9me3 to the *cenH* region (Fig. 6c). Remarkably, targeting of H3K9me3 to *cenH* was severely affected when *swi6Δ* was combined with *CCAAT^mut* or *crypticSS^mut* (Fig. 6c, d). Moreover, loss of any subunit of PhpC or Moc3 in *swi6Δ* cells yielded similar results for both H3K9me3 and H3K9me2 (Fig. 6c and Supplementary Fig. 7d). Consistent with our results showing that TFs are dispensable for Swi6[HP1]-dependent production of siRNAs mapping to *cenH* (Fig. 6a, b and Supplementary Fig. 7c), loss of these factors, *CCAAT^mut* or *crypticSS^mut* alone had no major impact on heterochromatin assembly in cells expressing functional Swi6[HP1] (Fig. 6c, d and Supplementary Fig. 7d).

To further validate our findings, we assessed the impact of combining *CCAAT^mut* with *swi6Δ* on the localization of the Clr4[Suv39h] complex. Raf2, a subunit of the Clr4[Suv39h] complex, is normally distributed throughout the silent *mat* region[13]. However, its localization was restricted to the *cenH* nucleation site in *swi6Δ* cells (Supplementary Fig. 7e). Consistent with the results observed for H3K9me3, loss of TFs or mutating *CCAAT* in *swi6Δ* cells drastically reduced the Raf2 loading at *cenH* (Supplementary Fig. 7e). Together, these results indicate that TF-mediated *cenH* transcription plays an important role in cryptic intron-dependent generation of siRNA and heterochromatin nucleation, which is unveiled upon loss of Swi6[HP1].

### TFs promote de novo establishment of a repressive heterochromatin domain

Swi6[HP1]-independent production of siRNAs may play a role in the de novo targeting of H3K9me and heterochromatin assembly. To investigate this, we studied whether PhpC binding to *cenH* is necessary for the initial formation of heterochromatin at the silent *mat* region using a sensitized reporter system. In *mat1-M* cells lacking a local silencer element (*REII*), the disruption of heterochromatin assembly leads to derepression of the *mat2-P* locus. This results in abnormal sporulation (haploid meiosis) due to the simultaneous expression of M (*mat1M*) and P (*mat2P*) mating-type information in haploid cells. Cells undergoing haploid meiosis exhibit dark staining upon exposure to iodine vapor, while WT colonies stain yellow. Additionally, the level of expression of *ura4^+* inserted near *mat2P* (*mat2P::ura4^+*) offers a further assessment of heterochromatic silencing[57].

To conduct the heterochromatin establishment assay, we cultured WT and *CCAAT^mut* cells, both carrying the *REIIΔ mat2P::ura4^+* reporter, in the presence of the histone deacetylase inhibitor trichostatin A (TSA). TSA is known to erase H3K9me marks and relieve gene silencing[36]. Following the removal of TSA, WT cells successfully reestablished silencing at the silent *mat* region (Fig. 7a). However, *CCAAT^mut* cells were deficient in reestablishing heterochromatic silencing, as indicated by the expression of *mat2P::ura4^+* and the dark iodine staining of colonies (Fig. 7a). This is consistent with lower levels of H3K9me3 in *CCAAT^mut* cells compared to WT cells (Fig. 7b). Conversely, both WT and *CCAAT^mut* cells cultured without TSA maintained reporter gene silencing, forming colonies that stained yellow when exposed to iodine vapor (Fig. 7a), consistent with similar levels of H3K9me3 (Fig. 6c).

Next, we utilized a fluorescence-based single-cell silencing assay to investigate the kinetics of de novo heterochromatin establishment in *CCAAT^mut* cells that were defective in TF binding at *cenH*. Replacing the *ura4^+* reporter with a *GFP* gene enables accurate measurement of the fraction of GFP-positive ("ON") cells after TSA washout. As expected, TSA treatment induced *GFP* expression in most cells in both WT and *CCAAT^mut* strains (Fig. 7c). Following TSA washout, WT cells reestablished *GFP* reporter silencing more rapidly than the *CCAAT^mut* mutants, which showed compromised reestablishment of heterochromatic silencing, as indicated by the significant number of cells that continued to express GFP, even after more than 20 generations (Fig. 7c, d). Similar results were observed in *CCAAT^mut moc3Δ* cells

(Supplementary Fig. 8a, b). It is noteworthy, however, that the proportion of *CCAAT^mut* cells with a silenced *mat* region increased with each generation in a stochastic manner, suggesting that heterochromatin eventually forms at this locus, likely through alternative nucleation mechanisms[36,58].

Together, these results emphasize the important role of TFs in promoting the establishment of heterochromatin. We find that TFs drive the transcription of cryptic intron-containing *cenH* transcripts, which are necessary for the Swi6[HP1]-independent production of siRNAs involved in targeting H3K9me and heterochromatin assembly.

## Discussion

Heterochromatin is generally repressive, preventing the underlying DNA sequences from being accessed by transcriptional machinery. Nonetheless, RNAPII transcription of heterochromatic repeat elements occurs and is necessary for recruiting heterochromatin assembly proteins to nucleation sites, such as through RNAi machinery[2,21–24,59]. Despite advances in understanding heterochromatin assembly pathways, exactly how heterochromatic repeats are transcribed remains to be fully elucidated. In this study, we demonstrate that two distinct transcription factors co-bind to specific sites within repeat elements in constitutive heterochromatin domains. Our analyses reveal that the binding of these transcription factors is crucial for transcribing RNAs that contain multiple cryptic introns, engaging RNAi machinery through spliceosome components, and triggering siRNA production for de novo heterochromatin assembly.

We show that in addition to a Zn-finger domain-containing protein Moc3, a trimeric transcription factor complex PhpC, related to the mammalian NF-Y[44,45], drives transcription of one of two strands of its target heterochromatic repeat elements. A recent study implicated Moc3 as a positive regulator of *dh* element transcription but could not detect its binding at the target repeats in wild-type cells[52]. This study suggested that TF binding is blocked once heterochromatin is assembled. Our results contrast with these previous findings. We show that both Moc3 and PhpC can access *cenH* and *dh* repeat elements, irrespective of the heterochromatin state. Indeed, comparable levels of both TFs are detected at these loci in WT and *clr4Δ* cells. Furthermore, we show that both TFs can infiltrate heterochromatin even in the absence of major HATs and chromatin remodelers. It is clear from these results that constitutive heterochromatin is not completely inaccessible to TFs that are required to promote the transcription of repeat elements.

How do TFs gain access to sequences within heterochromatin regions? Interestingly, the localization of PhpC and Moc3 to the heterochromatin region occurs in an interdependent manner, while this is not the case for binding to euchromatin regions. Loss of PhpC or mutating its binding site (*CCAAT*) dramatically impairs Moc3 binding to the *cenH* element. Similarly, we find that *moc3Δ* cells are defective in PhpC localization at heterochromatic repeats. These findings suggest that the collaborative action of these TFs is crucial for accessing heterochromatic regions. Interestingly, the PhpC subunits, Php3 and Php5, each possess an H2B/H2A-like HFD involved in histone-histone and histone-DNA interactions[46]. Considering that the human counterpart of PhpC, NF-Y, forms a nucleosome-like structure to gain chromatin accessibility[60–62], it is plausible that PhpC uses HFDs to displace nucleosomes to access the *CCAAT* box[63]. In this regard, we show that mutating conserved residues in the HFDs of Php3 or Php5 disrupts their localization to heterochromatin. TF heterochromatin accessibility is likely further facilitated by binding of Moc3 to DNA via its Zn-finger domain. In this sense, PhpC and Moc3 function as "pioneer transcription factors" to displace nucleosomes to access underlying DNA sequences[64]. Regardless of their binding mode to heterochromatic sequences, our analyses show that TFs act directly to promote repeat transcription, as opposed to causing indirect effects resulting from misregulation of their target gene expression. Indeed,

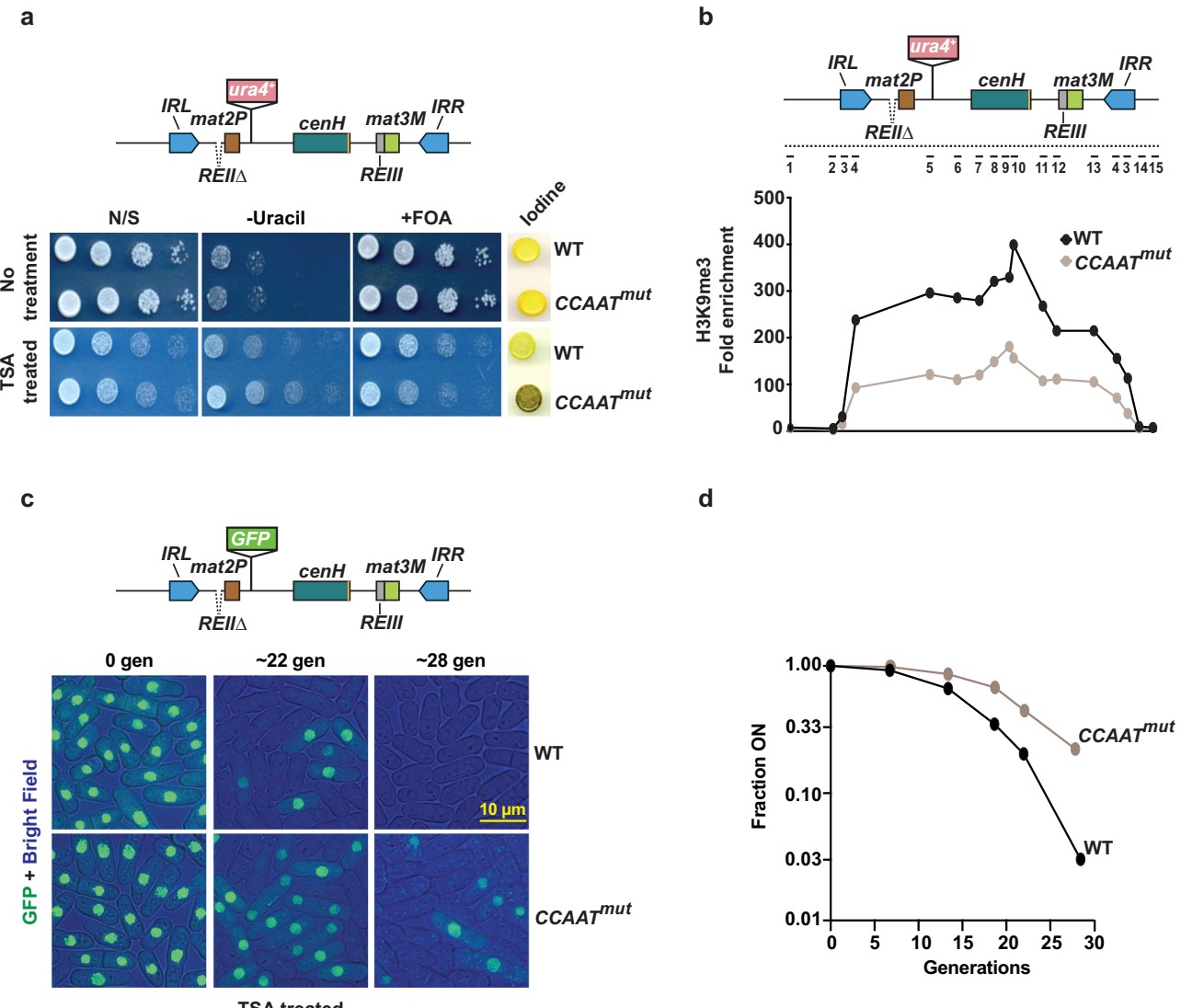

**Fig. 7 | PhpC contributes to de novo heterochromatin establishment. a** 10-fold serial dilution assay of wild-type (WT) and *CCAAT^mut* strains on the indicated medium before and after trichostatin A (TSA) treatment and washout. Iodine staining was performed on non-selective (N/S) plates. **b** ChIP-qPCR analysis of H3K9me3 enrichment at the silent *mat* locus in the indicated TSA-treated strains. Data is presented as the mean of 3 independent biological experiments. Numbered lines in the schematic indicate the location of the primers. **c** Representative live-cell images of WT and *CCAAT^mut* strains harboring the *mat2P::GFP* reporter at the indicated time points (generations) after TSA treatment and washout. **d** Quantification of the cell fraction in the "ON" state (e.g., GFP-positive) at the indicated time points is shown. N = 260–651 cells for each data point. Source data are provided as a Source Data file.

we show that mutating the *CCAAT* box, which affects both PhpC and Moc3 binding, impairs *cenH* transcription similarly to what is observed in the absence of these TFs.

Previous studies have shown that RNAPII transcription of the heterochromatic repeat elements *cenH* and *dh* occurs preferentially during the S-phase of the cell cycle[23,24]. This S-phase requirement for repeat transcription is dictated by the heterochromatin machinery involved in transcriptional silencing[2]. In cells defective in heterochromatin assembly, *cenH* and *dh* elements are transcribed throughout the cell cycle[23]. The restriction of heterochromatic repeat transcription to the S-phase contrasts with our findings that both PhpC and Moc3 can bind to heterochromatic repeats outside of the S-phase in wild-type cells. This suggests that heterochromatin obstructs additional steps necessary for RNAPII transcription beyond TF binding. Heterochromatin-associated HDACs, which are believed to prevent opening of chromatin by restricting access of chromatin remodelers to heterochromatin[65,66] and counteracting an anti-silencing factor Epe1[67], may prevent the assembly of active RNAPII transcriptional complexes,

thereby inhibiting transcription initiation and/or elongation. In this regard, the perturbation of heterochromatin during the S-phase, such as during DNA replication, may provide a "window of opportunity" for TFs to recruit RNAPII complexes and promote transcription of heterochromatic repeats.

Our findings reveal that the generation of *cenH* transcripts via TFs is crucial for producing siRNAs independent of Swi6^HP1 and for the de novo targeting of H3K9me by Clr4^Suv39h. In cells lacking Swi6^HP1, loss of PhpC, Moc3, or mutation of the *CCAAT* box, which impair the transcription of the *cenH* bottom strand, significantly impede siRNA production. The bottom strand *cenH* transcript contains multiple cryptic introns, and their processing into siRNAs by RNAi factors requires splicing machinery. Importantly, we show that mutating cryptic intron splice sites in cells lacking Swi6^HP1 results in drastic reduction in siRNA production. These observations, together with previous studies[2], suggest a model for the initial generation of siRNAs that trigger de novo heterochromatin assembly at target repeat loci. The spliceosome, bound to TF-driven cryptic intron-containing transcripts, likely

provides specificity for recruiting RNAi factors to repeat transcripts for initial siRNA production (Fig. 8). In this regard, it has been shown that stalling of the spliceosome at cryptic introns recruits RDRC[56,68], likely in collaboration with a homolog of mammalian Ars2 (named Pir2 in *S. pombe*)[54] to generate double-stranded RNA substrates for Dcr1. The siRNAs produced through this process are loaded onto Ago1-containing RITS, which facilitates the recruitment of Clr4[Suv39h] to methylate H3K9. Once heterochromatin is established, TFs remain associated with repeat elements for continuous production of siRNAs. However, Swi6[HP1] bound to H3K9me can independently recruit RDRC to further promote the RNAi-mediated processing of bidirectional repeat transcripts and amplify siRNA production[31–33]. This model explains why mutating TFs or their recruitment sites at *cenH* in cells with functional Swi6[HP1] does not significantly impact siRNA production and heterochromatin assembly.

Our ability to specifically block *cenH* bottom-strand transcription has provided crucial insight into how cells initiate siRNA production to nucleate heterochromatin via mechanisms involving cryptic introns and the splicing machinery. Other mechanisms, such as the generation of Dcr1-independent but Ago1-dependent small RNAs, referred to as primal RNAs, might also contribute[69]. However, considering the studies showing that H3K9me levels are similar in *ago1Δ* and *dcr1Δ* cells, and even in *ago1Δ dcr1Δ* double mutant cells[70,71], the role of primal RNAs is likely only minor.

The RNA-based targeting of heterochromatin is widely conserved across various systems. RNAi, an ancient defense and regulatory mechanism, plays a crucial role in forming heterochromatin at repetitive DNA elements and other loci in various organisms, including *C. elegans* and plants[72–75]. This suggests the potential existence of factors analogous to the TFs described here, which may infiltrate repressive chromatin domains to mediate transcription at RNAi target loci, thus maintaining a steady pool of siRNAs via mechanisms that also require splicing factors[68,76–78]. Indeed, several transcription factors have been identified as binding to heterochromatic repeats in higher eukaryotes[79–82]. Although these TFs generally repress transcription, it is plausible that their binding promotes transcription of heterochromatic repeats under specific conditions to facilitate RNA-mediated heterochromatin formation. In this regard, we note that multiple pathways utilize RNAPII transcription to nucleate heterochromatin[2,83–86]. Besides the RNAi machinery, evolutionarily conserved nuclear RNA processing factors, including Enhancer of Rudimentary homolog (ERH), are required for H3K9me3 and silencing of both repetitive DNA elements and lineage-specific genes[2,15,83,84,87].

Future studies will investigate whether TFs capable of infiltrating repressive chromatin domains, including "pioneer transcription factors", contribute to maintaining heterochromatin domains through RNA-based mechanisms.

## Methods

### Strains and growth conditions
Strains were created through genetic crosses or a PCR-based method. The addition of epitope tags or gene deletions was confirmed using gene-specific PCR primers. The expression and nuclear localization of GFP-tagged transcription factors were confirmed through live-cell imaging.

The construction of the strain harboring GFP at the *Xba*I site next to *mat2-P(Bg-Bs)* was previously described[65]. The *CCAAT[mut]* and *crypticSS[mut]* strains were generated using CRISPR, and derived strains were generated by a genetic cross. The strains were cultured in rich yeast extract plus adenine (YEA) media using standard protocols unless otherwise specified. Oligonucleotides and strains used in this study are listed in Supplementary Data 5.

### Chromatin immunoprecipitation (ChIP)
The strains with epitope tags were grown until $OD_{600}$ 0.5. The cultures were then shifted to 18 °C for 2 hours and cross-linked with 3% paraformaldehyde (Sigma, Cat# P6148) for 30 minutes at 18 °C. Cells were washed with 1x PBS and crosslinked with dimethyl adipimidate dihydrochloride (DMA; Sigma Cat#285625) for 45 minutes at room temperature. Subsequently, the cells were washed again with 1x PBS and lysed with zirconia beads plus lysis buffer (50 mM HEPES/KOH pH 7.5, 140 mM NaCl, 1 mM EDTA, 1% Triton X-100, 0.1% deoxycholate, and protease inhibitors, Roche, Cat#183617001). The cell lysates were sheared using a Bioruptor-300 (Diagenode) to an approximate size of 300–600 base pairs. 3 µg of anti-GFP antibody (Abcam, Cat#ab290) was used for GFP immunoprecipitation. For RNA polymerase II immunoprecipitation, 25 µl of anti-Pol II antibody (8WG16, Santa Cruz Biotechnology, sc56767) was used. For Raf2-Myc immunoprecipitation, 25 µl of c-Myc antibody (Santa Cruz Biotechnology, Cat#9E10) was used. For H3K9me immunoprecipitation, 2 µl of anti-H3K9me2 (Abcam, Cat#ab115159) and anti-H3K9me3 (Abcam, Cat#ab8898) antibodies were used. Chromatin extracts were incubated with antibodies overnight at 4 °C, and an equal volume (25 µl) of Protein A plus (Pierce, Ref#22811) or Protein G (Invitrogen, Ref#15920010) agarose beads were added for four hours at 4 °C. For FLAG immunoprecipitation, 50 µl of anti-FLAG M2 affinity gel (Sigma, Cat#A2220) was directly

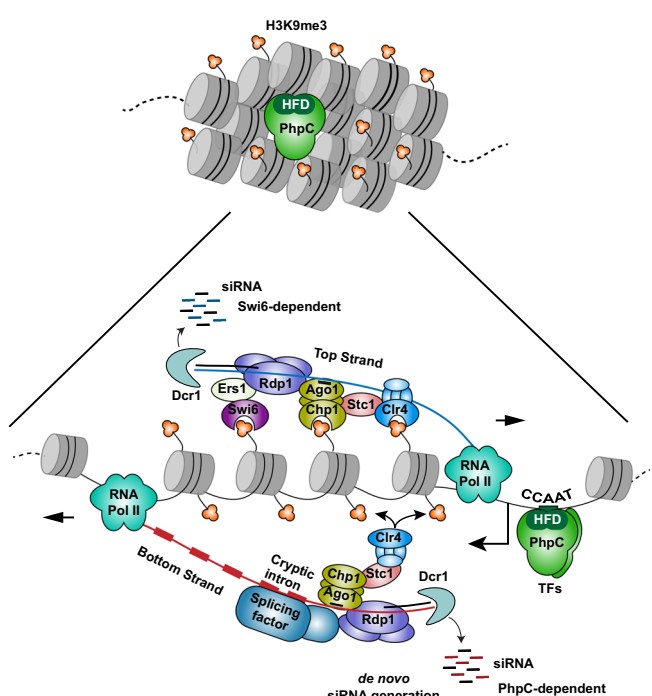

**Heterochromatin**

**Fig. 8 | Model showing the role of the histone fold domain-containing PhpC[NF-Y] in heterochromatic repeat transcription and siRNA generation.** The top panel illustrates how the histone fold domain (HFD) containing the trimeric TF complex PhpC, which recognizes underlying *CCAAT* boxes at heterochromatic regions such as *cenH*, binds to heterochromatic repeats in cooperation with Moc3. The bottom panel depicts how PhpC facilitates the transcription of the *cenH* bottom strand containing multiple cryptic introns (depicted as red rectangles). These introns likely stall the spliceosome that then recruits the RNAi machinery, particularly the RNA-dependent RNA polymerase Rdp1-containing protein complex RDRC, initiating siRNA production. The siRNAs are loaded onto the Ago1-containing RITS complex. Together with the protein Stc1, RITS recruits the Clr4[Suv39h] methyltransferase, promoting H3K9me and leading to de novo establishment of heterochromatin, independently of Swi6[HP1]. Once H3K9me is established, Swi6[HP1] binds and also recruits the RNAi machinery via Ers1, maintaining a pool of siRNAs that further facilitate heterochromatin assembly. Consequently, the simultaneous disruption of both pathways results in the loss of siRNAs and H3K9me.

incubated with chromatin extract for 4 hours at 4 °C. The beads were then washed twice with lysis buffer, once with lysis buffer containing 0.5 M NaCl, once with wash buffer (10 mM Tris-HCl pH8, 0.25 M LiCl, 1 mM EDTA, 0.5% NP-40, 0.5% DOC), and once in TE buffer, pH 8. The chromatin–antibody complex was eluted using TES buffer (50 mM Tris-HCl pH 8, 10 mM EDTA, 1% SDS) and, along with the whole-cell extract (WCE) input, was de-crosslinked by heating to 65 °C overnight and purified using MinElute Spin Columns (Qiagen Cat# 28004). For qPCR analyses, iTaq Universal SYBR Green Supermix (Bio-Rad, Cat# 1725120) was used. Delta-delta Ct normalization used *leu1* as the reference gene in all cases unless indicated otherwise. The fold enrichment was further normalized to tagged/untagged in all cases unless indicated otherwise. Oligonucleotides used for ChIP-qPCR are listed in Supplementary Data 5.

### Chromatin immunoprecipitation-sequencing (ChIP-seq)
Sequencing libraries were generated using NEBNext Ultra II DNA library prep kit for Illumina (Illumina, Cat# E7645) according to the manufacturer's protocol. The library size was analyzed using an Agilent 4200 Tape Station system (Agilent). Samples were multiplexed, and single-end reads were sequenced on the Illumina MiSeq platform using the MiSeq reagent kit V3 according to the manufacturer's protocol.

### Protein purification and mass spectrometry
GFP-tagged Php5 was purified from 2 liters of cell culture. Cells expressing Php5-GFP or the untagged control were cultured at 30 °C. The cells were washed with water and then flash-frozen in liquid nitrogen. The cell pellets were thawed on ice and ground with glass beads in a Pulverisette 6 system (Labsynergy) in lysis buffer (20 mM HEPES–KOH pH 7.6, 20% glycerol, 500 mM NaCl, 2 mM MgCl2, 150 mM KCl, 0.1% IGEPAL, 1 mM dithiothreitol (DTT), 1 mM EDTA, Roche complete protease inhibitors cocktail and 1 mM PMSF). The lysate was then cleared by centrifugation at 27,000 g for 1 hour, and the supernatant was incubated with anti-GFP agarose beads (GFP-Trap Chromotek, Cat#AB_2631357) for 2 hours at 4 °C. The beads were thoroughly washed in BAC150 (20 mM HEPES–KOH pH 7.6, 20% glycerol, 2 mM MgCl2, 1 mM EDTA, 150 mM KCl, 0.1% IGEPAL, Roche complete protease inhibitors cocktail, and 1 mM PMSF) and then eluted with 200 µl of 0.2 M glycine (pH-2.5). 1/10 of the eluted proteins were probed with an anti-GFP antibody (Roche) to confirm the efficiency of the pull-down (Uncropped blots are shown in the Source Data file), and the rest of the eluted proteins were precipitated with 10% TCA and resuspended in the sample buffer. The proteins were resolved on a 4%–12% Bis-Tris Gel (Invitrogen, Cat#NP0321BOX) and visualized using SimplyBlue SafeStain (Invitrogen, Cat#465034). Protein bands were excised from the gel and subjected to mass spectrometry. All purification and mass spectrometry experiments were repeated twice. The excised gel bands were subjected to in-gel trypsin digestion at a 20 ng/µl concentration for effective peptide extraction. The samples were desalted using Pierce C18 spin columns (Thermo Fisher Scientific), dried, and resuspended in 0.1% trifluoroacetic acid before loading onto an Acclaim PepMap 100 C18 LC column (Thermo Fisher Scientific). This process was carried out with a Thermo Easy nLC 1000 LC system connected to a Q Exactive HF mass spectrometer (Thermo Fisher Scientific). Peptide elution occurred through a 5% to 36% acetonitrile gradient with 0.1% formic acid over 56 minutes at a 300 nL/min flow rate. The QEHF performed MS1 scans in the orbitrap at a resolution of 60,000, with a maximum injection time of 120 ms and an AGC target of 1e6. For MS2 scans, a normalized collision energy of 27 was applied at a resolution of 15,000, with a maximum injection time of 50 ms and an AGC target of 2e5. The raw MS data was analyzed with Proteome Discoverer 2.2 and SEQUEST HT software, targeting the UniProt *S. pombe* proteome database from the European Bioinformatics Institute (https://www.uniprot.org/proteomes/UP000002485). The parent ion mass tolerance was 10 ppm, while the fragment ion

mass tolerance was set to 0.02 Da. The minimum peptide length was established at 6 amino acids, allowing for a maximum of two missed cleavages.

### RNA isolation and cDNA preparation
Cells were cultured in a YEA medium under normal growth conditions and then transferred to 18 °C for 2 hours before total RNA isolation. The RNA was isolated by incubating yeast cells in hot phenol (pH 5.2) at 65 °C for 15 minutes, followed by three additional phenol-chloroform extractions[88]. The RNA was precipitated using sodium acetate-ethanol and quantified using NanoDrop (Thermo). Approximately 50 µg of total RNA was treated with Turbo DNase I (Invitrogen, Cat# AM2238), and about 1 µg of total RNA was used for cDNA preparation using Oligo (dT) and hexamer primers following the manufacturer's protocol.

### qPCR and RT-PCR
Primers specific to *cenH* were used to assess expression (see Supplemental Data 5). We performed qPCR using iTaq Universal SYBR Green Supermix (Bio-Rad Cat# 1725120) on the QuantStudio3 platform (Thermo Fisher Scientific). For RT-PCR, we used Taq DNA polymerase (NEB, Cat#M0273L) and followed the manufacturer's instructions. The *leu1* gene served as the endogenous control for normalizing the transcript levels. Uncropped gels are shown in the Source Data file.

### RNA-seq library preparation
Cells were cultured in YEA medium and transferred to 18 °C for 2 h before isolating total RNA using the hot phenol method. The RNA quality was checked (RIN > 8), and approximately 350 ng of total RNA was used for rRNA removal with the TruSeq Standard Total RNA LP kit (Illumina, Ref#15032615). Libraries were prepared using the NEBNext Ultra Directional RNA Library Prep kit (NEB, Cat#E7760) and sequenced on the NextSeq500 platform.

### Small RNA-seq library preparation
Small RNA was purified from 4 OD$_{595}$ units of log phase cells using the MasterPure Yeast RNA Purification Kit (Lucigen Cat# MPY03100). Small RNAs (21-25nt) were excised after denaturing electrophoresis, and then ethanol precipitated and resuspended in DEPC-treated water. Adaptor ligation and PCR amplification were performed using the NEBNext Small RNA Library Prep Set for Illumina (NEB, Cat# E7300S). The final library was sequenced on the MiSeq platform.

### TSA treatment and dilution assay
Cells were initially cultured in YEA media and subsequently treated with TSA (MedChemExpress, Cat#HY-15144, used at a final concentration of 35 µg/ml) for 60 hours. After washing out TSA, the cells were grown in PMG-*ura* media for 24 hours, then diluted to -0.005 OD$_{600}$ into YEA media and incubated again for another 60 hours. Both untreated and TSA-treated cells were serially diluted and spotted on appropriate PMG plates. H3K9me3 ChIP-qPCR was performed as previously described. Data from three independent biological replicates are shown in the figures.

### Microscopy analysis of TSA-treated cells
Cells were initially grown in YEA media and then treated with TSA for 24 hours (approximately six generations) using the same conditions as above. After treatment, cells were washed with water and cultured in YEA liquid media at 26 °C for 4 days. Daily dilution to 0.025OD$_{600}$ in fresh YEA media was performed to prevent the cells from entering the stationary phase. Under these conditions, the doubling time was approximately 3 hours. For imaging, the cells were mounted in 2% agarose. Twenty 0.35-µm z-sections were captured for GFP fluorescence using a Delta Vision Elite microscope (Leica) with a 100×1.35 NA oil lens (Olympus). Images were deconvolved and projected into 2D

maximum-intensity images using SoftWorx (Leica). Analysis was done using ImageJ-based Fiji software.

## Multiple sequence alignments
Protein sequences were downloaded from the respective database and aligned using Jalview software.

## Cryptic-intron identification at *cenH*
*cenH* shares 96% homology with the *cen2 dh* element (*SPRPTCENB.4*). *cen2 dh* has six cryptic introns, which are detectable in wild-type (two cryptic introns) and *rrp6Δ ago1Δ* double mutant cells (four cryptic introns). We carefully searched for these intron splice sites at *cenH* and identified five putative intronic sequences. We then mutated the 5′ and 3′ splice sites, including the branch point; additionally, we also mutated a few potential splice sites at *cenH* loci. These mutations were confirmed by Sanger sequencing.

## Statistics and reproducibility
The statistical significance of qPCR results was assessed using two-tailed paired t-tests. RT-PCR experiments were performed at least twice, yielding similar results, as shown in Fig. 5a and c. All fluorescence images in Figs. 4h, 7c, and Supplementary Figs. 2b, 3e, 5c, and 8a are from over 200 yeast cells.

## Data processing and computational analysis
**ChIP-seq data analysis.** Single-end short reads were quality trimmed with fastp[89] and aligned using the BWA aligner[90] to the *S. pombe* V2 reference sequence[91] in which the Pombase provided 20 kb *mat* contig was replaced with the sequence of the full 40 kb *mat* region (MAT). In addition, both the *mat* region on chromosome 2 (chrII 2,109,748–2,138,781), and the *mat1M* locus on the 40 kb MAT contig (MAT 4,489–5,615) were masked from alignment. Bedgraphs of ChIP enrichment were produced using the MACS2[92] 'callpeaks' function to make broad calls with options '–nomodel–extsize 147', followed by the MACS2 'bdgcmp' function to compute fold enrichment (FE). Bedgraphs of FE of the IP over input divided by the FE of the untagged control over untagged input, referred to hereafter as 'FE bedgraphs', were used in downstream analysis. These bedgraphs were converted to a more convenient fixed-interval format by taking the mean signal over non-overlapping 10 bp windows.

**TF site identification.** A library of TF binding sites was taken from the 'summits' BED files generated for ChIPs of 16 TFs in WT backgrounds by MACS2 when run as described above. Each site in this collective set was then expanded by 25 bp symmetrically to produce a library of sites of uniform 50 bp width. Overlapping sites were subsequently merged to yield the final set of 2273 unique transcription factor binding sites that were used in the downstream analysis.

After site identification, the signal strength for each TF at each site was determined from the FE bedgraphs. Because both the width and central tendency of site occupancy varied considerably among the 16 TFs, the TF signal strength per site was taken as the maximum signal over an extended interval of 1 kb straddling the site center. Signal strengths per site for TFs in mutant backgrounds were calculated similarly.

**RNA-seq data analysis.** Single-ended short reads from RNA-seq experiments were quality-trimmed using fastp[89] and aligned using the STAR aligner[93]. Variable interval bedgraphs, including both uniquely and multi-mapping reads and normalized to counts per million mapped reads, were generated by STAR and were further processed to produce 10 bp fixed interval versions in the manner described above for ChIP-Seq.

**Visualization of genomic alignments.** The Integrative Genomics Viewer[94] (v.2.13.1) was used to visualize bedgraphs of NGS alignments. The Integrated Genome Browser[95] was used to visualize small RNA SGR alignments.

**Site-centered heat maps.** ChIP signals for each of the 16 TFs in WT backgrounds were plotted across the subset of the 2273 sites originally assigned to the TF in the respective MACS 'summits' BED file. Signals were taken from FE bedgraphs and signal density was plotted across sites using a combination of the computeMatrix and plotHeatmap programs of the Deeptools suite[96].

Hierarchical clustering of the 16 TFs on the basis of the vector of signal strength at each of the 2273 TF sites was performed by applying the Ward algorithm to the Euclidian distances between vectors as implemented in the Orange data mining suite[97]. Euler diagrams were produced using the 'eulerr' R package (https://cran.rproject.org/web/packages/eulerr/index.html), and the significance of site set overlap was assessed using the hypergeometric distribution as implemented in the 'SuperExactTest' R package (https://www.r-project.org). Heatmaps were produced using the 'ComplexHeatmap' R package[98]. Gene enrichments were computed using the Orange bioinformatics package.

**Small RNA-seq data analysis.** Single-ended short reads were quality trimmed with fastp[89] and aligned to the *S. pombe* reference using Novoalign (www.novocraft.com/products/novoalign). SGR files, whose format allows the simultaneous display of read counts from both strands, were subsequently created from the Novoalign-generated BAM files using custom scripts (https://doi.org/10.5281/zenodo.14285969). SGRs were either filtered to remove multi-mapping reads or left unfiltered to include multi-mapping reads and subsequently normalized to counts per 10 million mapped reads (CP10M).

## Reporting summary
Further information on research design is available in the Nature Portfolio Reporting Summary linked to this article.

## Data availability
The datasets are available on the NCBI Gene Expression Omnibus (Accession number GSE269096). The mass spectrometry data are available on MassIVE (Accession number MSV000095108). Source data are provided with this paper.

## Code availability
The code used in this study is publicly available at Zenodo (https://doi.org/10.5281/zenodo.14285969).

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

## Acknowledgements

We thank Jemima Barrowman for her valuable help editing the manuscript and other members of the Grewal lab for their helpful discussions. We thank Gobi Thillainadesan for contributing to the SGR conversion script. This work was supported by the Intramural Research Program of the National Institutes of Health and National Cancer Institute (ZIA BC 010523 and ZIA BC 011208 grants to S.G.). Bioinformatics analysis in this study used the Helix and Biowulf Linux cluster at the National Institutes of Health.

## Author contributions

S.I.S.G. and M.K.S. designed the study. M.K.S., H.D.F., P.N., A.A., D.V., S.J., J.D., M.O., and T.A. performed experiments. M.K.S., J.D., and S.J. constructed the strains used in this study. P.N. constructed *CCAAT^mut* strains. H.D.F. performed live-cell imaging. A.A. analyzed cryptic introns SS site and constructed the *crypticSS^mut* strain. D.V. and A.A. performed small RNA-seq experiments. M.K.S. performed all other experiments. M.O. and T.A. performed mass spectrometry analysis. D.W. performed bioinformatic analyses. M.K.S. prepared the data figures, and M.K.S., H.D.F., and S.I.S.G. wrote the manuscript with input from coauthors.

## Funding

## Competing interests

The authors declare no competing interests.
