## [Peer Review file · Nature Communications]

PhpC^{NF-Y} transcription factor infiltrates heterochromatin to generate cryptic intron-containing transcripts crucial for small RNA production

Corresponding Author: Dr Shiv Grewal

Version 0:

Reviewer comments:

Reviewer #1

(Remarks to the Author)

Review of "PhpC^{NF-Y} transcription factor infiltrates repressive heterochromatin to generate transcripts crucial for spliceosome-dependent small RNA production" by Srivastav et al

The assembly of heterochromatic structures is essential for silencing repetitive DNA elements and developmental genes. Previous studies using *S. pombe* and other systems have shown that transcription of heterochromatin target loci, including repetitive elements, is required for nucleating heterochromatin via an RNAi-mediated mechanism. However, the transcription mechanism of these repressive heterochromatin-coated repeat elements was unclear. This study demonstrates that heterochromatin is not entirely inaccessible to transcription factors (TFs). The authors identify a trimeric TF complex, PhpC—related to the mammalian NF-Y complex—as critical for the transcription of heterochromatic repeats. Their results show that PhpC collaborates with a Zn-finger-containing TF to bind to repeat promoters with CCAAT boxes. Notably, this TF binding occurs regardless of heterochromatin status or cell cycle phase and is essential for repeat transcription. By mutating the TFs or the CCAAT binding site, the authors demonstrate that TFs directly promote repeat transcription. Additionally, the study reveals that TF-driven transcription is crucial for generating primary siRNAs through the RNAi machinery. Transcripts generated by TFs that contain cryptic introns are processed through a spliceosome-dependent pathway to produce siRNAs. The functional significance of this pathway is demonstrated by showing that the TF-mediated transcription pathway is vital for heterochromatin nucleation.

This elegant study addresses two longstanding questions. First, it identifies transcription factors capable of binding to repeat elements within repressive heterochromatin, thereby uncovering a mechanism by which heterochromatic repeats are transcribed. Second, it elucidates the primary cascade for siRNA generation by the RNAi machinery, which nucleates heterochromatin. The conclusions are robustly supported by the data. I have only minor suggestions for revision.

1. The binding of transcription factors (TFs) to chromatin often requires histone-modifying activities such as histone acetyltransferases (HATs). This raises a pertinent question about whether a HAT enzyme might facilitate TF localization to heterochromatic repeats. The authors could explore this possibility, specifically focusing on the major HAT activities in *S. pombe*.
2. There is ongoing debate regarding whether TF accessibility to repressive chromatin can occur without nucleosome remodeling activities. Considering that the *S. pombe* counterpart of PhpC, the NF-Y complex in mammals, is believed to act as a "pioneer factor," it would be valuable to investigate if nucleosome remodelers such as Ino80 or RSC (which co-purified with PhpC, as shown in Supplemental Figure 2a) are required for TF localization to heterochromatic repeats.
3. Based on Figure 1a, there are two distinct siRNA clusters at the centromere. Only one of these clusters (the dh element) shows enrichment of PhpC components. Do the authors have any hypotheses about which TFs might be responsible for transcription at the other siRNA cluster corresponding to the repeat labeled dg element?
4. A notable category among loci showing PhpC enrichment is long non-coding RNAs (lncRNAs). This finding is significant considering that PhpC is involved in the transcription of lncRNAs at heterochromatic regions. This raises the question of

whether this TF complex plays a general role in the transcription of lncRNAs. The authors should consider exploring, or at least discussing, this possibility.

Reviewer #2

(Remarks to the Author)

This is an interesting and important study that adds significantly to the field (regulation and transcription of repetitive sequences in heterochromatin) and will have a noticeable impact. The manuscript is well-written, clear, and easy to follow. However, there are important points that need to be addressed (major revision).

1. Results, Paragraph 1: The manuscript mentions 96% homology between cenH and the dg and dh repeats. I suggest adding a sentence that explains why ChIP-seq does not define whether the binding occurs at both or only one of these locations. A diagram that clarifies the position of the primers used for qPCR would also be important.
2. Results, Paragraph 2: When discussing the genome-wide distribution patterns of these TFs, it would be better to clarify whether the clustering was performed using the ChIP-seq signal or the binding sites. Additionally, the sentence "Gene Ontology analysis revealed that loci targeted by..." is unclear—specify whether the genes used for the GO analysis have binding sites in the gene body or in the promoter.
3. Results, Paragraph 3: In the figure showing the mass spectrometry data, it is important to display some "background" purified peptides that are positive in both the control and sample.
4. Results, Paragraph 3: Clarify how many base pairs near the summit of binding sites were used for the CCAAT analysis.
5. Results, Paragraph 7: The last sentence is unclear; please add a sentence to explain why the combination of swi6d with other TFs leads to a severe reduction of siRNAs.
6. Figures: In general, when a locus is shown, the author should indicate its size in terms of base pairs or its location in the genome.
7. Figure 4g: The data concerning the Top strand should be shown alongside panel g.
8. Supplementary Figure: Including a panel with the genomic PCRs that validate the different strains with the addition of tags or deletions could be useful.
9. Replicates: For RNA-seq, some ChIP-seq, and mass spectrometry, the authors performed only one replicate. In my opinion, this raises concerns about the reproducibility of the data, particularly when comparing signals across experiments. In the qPCR panels, the authors indicate three replicates (each bar contains three points). Are these technical or biological replicates?
10. Materials and Methods Section: The number of replicates for the different experiments needs to be specified.

Reviewer #4

(Remarks to the Author)

Srivastav et al. present an elegant and insightful study elucidating the molecular mechanisms underlying the production of siRNAs from CenH heterochromatin loci in *Schizosaccharomyces pombe*, a crucial process for the establishment and propagation of heterochromatin. The authors' identification and characterization of the roles of the trimeric transcriptional factor complex PhpC and Moc3 in binding to a specific motif within the dh region of the CenH heterochromatic locus is a noteworthy achievement. Their well-designed genetic and molecular biology experiments have convincingly demonstrated the crucial role of this transcription factor binding in generating a transcript with cryptic introns, which is subsequently recognized by the splicing machinery and recruits the RNA-dependent RNA polymerase (RdRP) complex to produce siRNAs. This study is a significant step forward in understanding the role of transcription factor binding in initiating the de novo establishment of H3K9me.

Additionally, the study reveals that the binding of these transcription factors persists throughout the cell cycle, even in the context of heterochromatin, offering new insights into the accessibility of heterochromatin by transcription factor complexes and RNA polymerase II machinery. The manuscript is well-written, and the experiments are thoughtfully executed, providing comprehensive support for the authors' conclusions. I have no major concerns but offer some minor comments for the authors to consider before publication.

Minor Comments:

1. The authors report that these transcription factors are specifically required for generating the bottom transcript at the CenH locus and that deletion of these factors is sufficient to abolish swi6-independent siRNA production. This suggests that a dsRNA is produced by RdRP using only the bottom transcript. However, the role of the top transcript remains unclear. Is the top transcript required for siRNA-induced heterochromatin formation, perhaps contributing to swi6-dependent siRNAs? If so, is the siRNA generation from the top transcript RdRP-dependent, or could it occur via pairing the two transcripts? Clarifying this point would enhance the understanding of the transcriptional dynamics at this locus.
2. The authors demonstrate the binding of the transcription factors to both CenH and euchromatic regions. It would be helpful to quantify and compare the binding affinity of these factors to CenH versus euchromatic regions. Is the binding to CenH as abundant as to euchromatic regions, or is there a differential affinity that might contribute to the specificity of heterochromatin formation?
3. The role of cryptic introns in heterochromatin establishment raises intriguing questions. Are these cryptic introns necessary for the formation of H3K9me domains? Could this mechanism discriminate between euchromatic and heterochromatic loci despite the binding of the same transcription factors? The authors might consider discussing whether

the presence of cryptic introns is a determining factor for heterochromatin formation or if other downstream mechanisms play a role in distinguishing between these two chromatin states. Adding a few sentences to the discussion on this point would provide valuable context.

4. To enhance clarity and impact, the authors could consider moving Supplementary Figure 4f closer to Figure 4g. This would emphasize that only the bottom transcript is affected and help visually connect this key finding.

Version 1:

Reviewer comments:

Reviewer #1

(Remarks to the Author)

The authors have addressed all of my concerns and the manuscript appears ready for publication.

Reviewer #2

(Remarks to the Author)

With the new results, the work is robust. Additionally, the inclusion of replicates strengthens the conclusions. I believe the paper will be published.

Minor comment:

For Fig. 4e–h and Supplementary Fig. 5, I suggest adding a quantification of the effect. This is particularly important for the mutant of Php5, as the mutation might completely abolish the binding of the trimer to DNA.

Reviewer #4

(Remarks to the Author)

The authors have successfully addressed all my concerns. The paper is suitable for publication in its current format.

Response to reviewers comments

We are most grateful to the reviewers for their valuable feedback and for supporting the publication of our work. Following the helpful comments from the reviewers and the editorial recommendations, we have performed additional experiments and have modified the text as recommended. Our revisions address all the concerns raised by the reviewers and further strengthen the main conclusions of our study.

Some of the changes that we have made to the manuscript are listed below.

- (a) As requested by reviewer #1, we have assessed the roles of histone acetyltransferases (HATs) and chromatin remodelers on PhpC transcription factor (TF) binding to heterochromatic repeats. Based on our results, the loss of major HAT activities and remodelers has only minimal impact on TF localization to heterochromatin. These new data are presented in **Supplementary Fig. 4 d-i**.

However, in an exciting new finding, our results show that the conserved histone-fold domains (HFDs) of the PhpC subunits Php3 and Php5 are critical for the infiltration of repressive heterochromatin domains by this TF complex. We found that introducing point mutations of the conserved residues in the Php3 and Php5 HFDs abolishes PhpC binding to heterochromatic repeats, whereas binding of this TF complex to euchromatic regions is only mildly affected. These new data are presented in **Fig. 4 e-h** and **Supplementary Fig. 5**.

- (b) Although not requested by the reviewers, we further examined the mechanism responsible for PhpC-dependent siRNA generation from *cenH* heterochromatic repeat elements at the silent mating-type region. Specifically, we have investigated whether multiple cryptic introns that are present in the *cenH* transcript transcribed by the PhpC are indeed required for small RNA production. For this, we constructed a mutant strain in which 5' and 3' splice sites of all five cryptic introns in the *cenH* transcript are mutated. Remarkably, our results show that cryptic introns in the *cenH* transcript are required for the generation of Swi6/HP1-independent siRNAs. We further show that loss of siRNAs in cryptic intron mutant cells is linked to defects in heterochromatin assembly as detected using H3K9me ChIP. These new results further strengthen our conclusions and show that TF-driven cryptic intron-containing transcripts are pivotal for siRNA generation and heterochromatin nucleation. This new data is presented in **Fig. 6 b, d**.
- (c) We added a second set of biological replicates to the RNA-seq, ChIP-seq, and mass spectrometry datasets.

(d) We have included the requested clarifications in the Methods, Results, and Discussion sections.

Our detailed response to the referee's comments, indicated by italics, are described below (responses shown in blue):

Reviewer #1

*The assembly of heterochromatic structures is essential for silencing repetitive DNA elements and developmental genes. Previous studies using *S. pombe* and other systems have shown that transcription of heterochromatin target loci, including repetitive elements, is required for nucleating heterochromatin via an RNAi-mediated mechanism. However, the transcription mechanism of these repressive heterochromatin-coated repeat elements was unclear. This study demonstrates that heterochromatin is not entirely inaccessible to transcription factors (TFs). The authors identify a trimeric TF complex, PhpC—related to the mammalian NF-Y complex—as critical for the transcription of heterochromatic repeats. Their results show that PhpC collaborates with a Zn-finger-containing TF to bind to repeat promoters with CCAAT boxes. Notably, this TF binding occurs regardless of heterochromatin status or cell cycle phase and is essential for repeat transcription. By mutating the TFs or the CCAAT binding site, the authors demonstrate that TFs directly promote repeat transcription.*

Additionally, the study reveals that TF-driven transcription is crucial for generating primary siRNAs through the RNAi machinery. Transcripts generated by TFs that contain cryptic introns are processed through a spliceosome-dependent pathway to produce siRNAs. The functional significance of this pathway is demonstrated by showing that the TF-mediated transcription pathway is vital for heterochromatin nucleation.

This elegant study addresses two longstanding questions. First, it identifies transcription factors capable of binding to repeat elements within repressive heterochromatin, thereby uncovering a mechanism by which heterochromatic repeats are transcribed. Second, it elucidates the primary cascade for siRNA generation by the RNAi machinery, which nucleates heterochromatin. The conclusions are robustly supported by the data.

We are thankful to the reviewer for this positive feedback and for the insightful suggestions. We have addressed all concerns as follows.

*1. The binding of transcription factors (TFs) to chromatin often requires histone-modifying activities such as histone acetyltransferases (HATs). This raises a pertinent question about whether a HAT enzyme might facilitate TF localization to heterochromatic repeats. The authors could explore this possibility, specifically focusing on the major HAT activities in *S. pombe*.*

As suggested, we performed ChIP-qPCR to examine the binding of Moc3 and the PhpC subunit Php3 in cells lacking the two major histone acetyltransferases (HAT) Gcn5 and Mst2. Notably, we observed only a slight reduction in the binding of Php3 and Moc3 to the relevant heterochromatic regions *cenH* and *dh* upon loss of both HATs (**Supplementary Fig. 4d, e**). This finding suggests that TFs can bind to heterochromatic repeats independently of these HAT activities.

*2. There is ongoing debate regarding whether TF accessibility to repressive chromatin can occur without nucleosome remodeling activities. Considering that the *S. pombe* counterpart of PhpC, the NF-Y complex in mammals, is believed to act as a "pioneer factor," it would be valuable to investigate if nucleosome remodelers such as Ino80 or RSC (which co-purified with PhpC, as shown in Supplemental Figure 2a) are required for TF localization to heterochromatic repeats.*

We assayed the localization of Php3 and Moc3 in the remodeler mutants *ino80*^{K873A} (containing a mutation within a conserved ATP-binding domain), *rsc1Δ*, *snf22Δ*, *rsc1Δ ino80*^{K873A} and *rsc1Δ snf22Δ*. We found that removing these remodelers only slightly reduced TF binding (**Supplementary Fig. 4f-i**). These results indicate that TFs can access heterochromatic repeats independently of these remodeler activities. In this regard, it is possible that once TFs are bound to heterochromatin regions, they may recruit chromatin remodelers that further enhance TF binding.

3. Based on Figure 1a, there are two distinct siRNA clusters at the centromere. Only one of these clusters (the dh element) shows enrichment of PhpC components. Do the authors have any hypotheses about which TFs might be responsible for transcription at the other siRNA cluster corresponding to the repeat labeled dg element?

We mapped other TFs in addition to PhpC and Moc3 and found that none of them localized to *dg* repeat elements. Consequently, we are uncertain which TFs are responsible for *dg* repeat element transcription and siRNA production.

4. A notable category among loci showing PhpC enrichment is long non-coding RNAs (lncRNAs). This finding is significant considering that PhpC is involved in the transcription of lncRNAs at heterochromatic regions. This raises the question of whether this TF complex plays a general role in the transcription of lncRNAs. The authors should consider exploring, or at least discussing, this possibility.

We agree with the reviewer's observation that PhpC localizes to several lncRNAs. Our analysis indicates that these lncRNAs are associated with growth, stress, and developmental gene regulation, such as *sme2*, which is essential for meiotic progression (**Supplementary Fig. 2e**). This suggests that PhpC is not only required for repeat element transcription and *de novo* heterochromatin assembly,

but it may also regulate gene expression by affecting the generation of other lncRNAs. Further work is required to validate this hypothesis.

Reviewer #2

This is an interesting and important study that adds significantly to the field (regulation and transcription of repetitive sequences in heterochromatin) and will have a noticeable impact. The manuscript is well-written, clear, and easy to follow. However, there are important points that need to be addressed (major revision).

We are grateful to the reviewer for supporting the publication of our work and for providing helpful suggestions. Below are responses to specific comments by the reviewer.

1. Results, Paragraph 1: The manuscript mentions 96% homology between cenH and the dg and dh repeats. I suggest adding a sentence that explains why ChIP-seq does not define whether the binding occurs at both or only one of these locations. A diagram that clarifies the position of the primers used for qPCR would also be important.

We appreciate this suggestion, and we have further clarified this issue. The *cenH* peak region contains a mix of reads, with some reads that map uniquely and some that map equally well to pericentromeric *dg/dh*. As a result, determining the proportion of cross-mapping reads that come from *cenH* and not from *dg/dh* through ChIP-seq analysis is challenging. To address this, we utilized qPCR in which one of the primers uniquely binds upstream of *cenH*, as now depicted in **Fig. 1c**. Furthermore, note that altering two CCAAT boxes at *cenH* drastically reduced the localization of Php3 and Moc3 at *cenH* while leaving the pericentromeric *dh* unaffected. This suggests that the *cenH* and *dh* peaks are independent.

2. Results, Paragraph 2: When discussing the genome-wide distribution patterns of these TFs, it would be better to clarify whether the clustering was performed using the ChIP-seq signal or the binding sites. Additionally, the sentence "Gene Ontology analysis revealed that loci targeted by..." is unclear—specify whether the genes used for the GO analysis have binding sites in the gene body or in the promoter.

We thank the reviewer for noticing this. The hierarchical clustering of the TFs based on the vector of signal strength at each of the 2,273 TF sites was performed by applying the Ward algorithm to the Euclidian distances between vectors. This clarification is now included in the Methods and text (page 5, last paragraph). Gene Ontology (GO) analysis revealed that both Moc3 and Php5 bind to the promoters of genes linked to various pathways involved in energy production, such as NADH

metabolic processes, ATP generation, and glycolytic pathways. This clarification is now included in the text (page 6, first paragraph).

3. Results, Paragraph 3: In the figure showing the mass spectrometry data, it is important to display some “background” purified peptides that are positive in both the control and sample.

Indeed, we have also detected peptides in the control. Below is an example of histone protein peptides that appeared in both Php5-GFP and untagged pulldowns.

		Biological replicate 1			Biological replicate 2		
		% Coverage			% Coverage		
		Php5	No tag		Php5	No tag	
	Protein	GFP	Control		GFP	Control	
Histone	H2A	57	23		66	17	
	H4	57	31		57	10	
	H2A.Z	55	12		55	0	
	H2B	54	36		59	35	
	H3	33	7		29	0	

4. Results, Paragraph 3: Clarify how many base pairs near the summit of binding sites were used for the CCAAT analysis.

The estimated number of distinct, functional CCAAT-box motifs associated with a Php2/3/5 peak involved counting the exact matches to the motif on both strands within a range of plus or minus 500 base pairs from the peak center. Previous research suggested that when CCAAT motifs are separated by approximately 25 base pairs, only one motif is functional. Therefore, motifs within 25 base pairs of each other were considered as a single motif. This clarification is now included in the text (page 7, first paragraph).

5. Results, Paragraph 7: The last sentence is unclear; please add a sentence to explain why the combination of swi6d with other TFs leads to a severe reduction of siRNAs.

We have demonstrated that TFs are responsible for transcribing the bottom strand of *cenH*. When we combine *swi6Δ* (note that Swi6 is an HP1 family protein) with deletions of TFs or the *CCAAT^{mut}*, the production of siRNAs mapping to *cenH* is drastically reduced. This indicates that TF-driven transcription of the bottom strand of *cenH* is necessary for producing Swi6^{HP1}-independent siRNA (refer to Fig. 6a). This clarification is now included in the text (page 11, last paragraph).

Furthermore, our analysis revealed that TF-driven bottom-strand transcripts contain multiple cryptic introns. We mutated the cryptic intron splice sites (referred to as *crypticSS^{mut}*) by CRISPR-Cas9. Mutating the splice sites alone did not affect siRNA production from *cenH*. However, when combined

with *swi6Δ*, the mutation abolished siRNA production from *cenH* (Fig. 6b). This interesting observation mimics the effect of combining *swi6Δ* with deletions of TFs or a mutation in the spliceosome subunit *cwf10* (Supplementary Fig. 7c). Together, these findings indicate that TF-driven cryptic intron-containing bottom-strand *cenH* transcripts engage spliceosome machinery to generate siRNAs required for initial heterochromatin formation.

6. Figures: In general, when a locus is shown, the author should indicate its size in terms of base pairs or its location in the genome.

We appreciate the reviewer for pointing this out. The size of *cenH* is ~4.2 kb. The loci sizes and/or genomic coordinates are now indicated in each relevant figure panel.

7. Figure 4g: The data concerning the Top strand should be shown alongside panel g.

The top strands are now shown alongside the bottom strands (Fig. 5c, d).

8. Supplementary Figure: Including a panel with the genomic PCRs that validate the different strains with the addition of tags or deletions could be useful.

We use antibiotic resistance markers such as KanMX (resistance to G418) or NatMX (resistance to nourseothricin) to track gene tagging or deletion. In our lab, we routinely generate strains by crossing and tetrad dissection, followed by replica plating, and track their growth on selectable plates containing antibiotics.

9. Replicates: For RNA-seq, some ChIP-seq, and mass spectrometry, the authors performed only one replicate. In my opinion, this raises concerns about the reproducibility of the data, particularly when comparing signals across experiments. In the qPCR panels, the authors indicate three replicates (each bar contains three points). Are these technical or biological replicates?

We have completed the biological replicates for Php5-GFP IP/MS and have uploaded them to MassIVE (MSV000095108). We have included biological replicates for all our small RNA-seq and RNA-seq experiments, ensuring that we can confidently reproduce the data. We have also performed two biological replicates for all ChIP-seq experiments, except for those that were not used for genome-wide binding analyses, in which case we validated our findings by ChIP-qPCR of three independent biological replicates. The newly generated datasets can now be accessed at GEO Accession no GSE269096.

10. Materials and Methods Section: The number of replicates for the different experiments needs to be specified.

We have specified the number of replicates in each figure panel.

Reviewer #4

Srivastav et al. present an elegant and insightful study elucidating the molecular mechanisms underlying the production of siRNAs from CenH heterochromatin loci in Schizosaccharomyces pombe, a crucial process for the establishment and propagation of heterochromatin. The authors' identification and characterization of the roles of the trimeric transcriptional factor complex PhpC and Moc3 in binding to a specific motif within the dh region of the CenH heterochromatic locus is a noteworthy achievement. Their well-designed genetic and molecular biology experiments have convincingly demonstrated the crucial role of this transcription factor binding in generating a transcript with cryptic introns, which is subsequently recognized by the splicing machinery and recruits the RNA-dependent RNA polymerase (RdRP) complex to produce siRNAs. This study is a significant step forward in understanding the role of transcription factor binding in initiating the de novo establishment of H3K9me.

Additionally, the study reveals that the binding of these transcription factors persists throughout the cell cycle, even in the context of heterochromatin, offering new insights into the accessibility of heterochromatin by transcription factor complexes and RNA polymerase II machinery. The manuscript is well-written, and the experiments are thoughtfully executed, providing comprehensive support for the authors' conclusions. I have no major concerns but offer some minor comments for the authors to consider before publication.

We are thankful to the reviewer for supporting the publication of our work. The minor concerns of the reviewer have been addressed as follows:

Minor Comments:

1. The authors report that these transcription factors are specifically required for generating the bottom transcript at the cenH locus and that deletion of these factors is sufficient to abolish Swi6-independent siRNA production. This suggests that a dsRNA is produced by RdRP using only the bottom transcript. However, the role of the top transcript remains unclear. Is the top transcript required for siRNA-induced heterochromatin formation, perhaps contributing to swi6-dependent siRNAs?

We thank the reviewer for the suggestion. Although we have not tested this interesting possibility, we believe that transcription of the top strand must contribute to the formation of Swi6-dependent

siRNAs since these siRNAs are readily observed in cells devoid of PhpC or Moc3, in which transcription of the bottom strand is abrogated.

If so, is the siRNA generation from the top transcript RdRP-dependent, or could it occur via pairing the two transcripts? Clarifying this point would enhance the understanding of the transcriptional dynamics at this locus.

The contribution of the hypothetical pairing of both transcripts cannot be completely ruled out, although it is well established that RdRP is essential for siRNA generation in *S. pombe*. Based on our results with the spliceosome and cryptic intron mutants, we speculate that the spliceosome binds to the cryptic introns in the bottom strand. This event would recruit RdRP to generate double-stranded RNAs (dsRNA), which in turn are known to be processed by RNAi, leading production of siRNAs, H3K9me3 nucleation and Swi6 binding. Since Swi6 can recruit RdRP through Ers1, it is very plausible that Swi6 bound to H3K9me triggers generation of dsRNA and siRNA production from the top strand. Details are provided in Fig. 8 (model).

2. The authors demonstrate the binding of the transcription factors to both CenH and euchromatic regions. It would be helpful to quantify and compare the binding affinity of these factors to CenH versus euchromatic regions. Is the binding to CenH as abundant as to euchromatic regions or is there a differential affinity that might contribute to the specificity of heterochromatin formation?

We have assessed the binding affinity of TFs in *cenH*, centromeres, and euchromatin. Comparisons are shown below, in which we observe no gross changes in binding affinity, except for euchromatin vs. centromeres for Moc3. These results suggest that the specificity of heterochromatin formation is not due to the differing binding affinities of these TFs.

3. *The role of cryptic introns in heterochromatin establishment raises intriguing questions. Are these cryptic introns necessary for the formation of H3K9me domains? Could this mechanism discriminate between euchromatic and heterochromatic loci despite the binding of the same transcription factors? The authors might consider discussing whether the presence of cryptic introns is a determining factor for heterochromatin formation or if other downstream mechanisms play a role in distinguishing between these two chromatin states. Adding a few sentences to the discussion on this point would provide valuable context.*

Thank you for raising this important point. As we described in the response to Reviewer 2, our analysis revealed that TF-driven bottom-strand transcripts contain multiple cryptic introns. We have now mutated the cryptic intron splice sites (referred to as *crypticSS^{mut}*) by CRISPR-Cas9. Mutating cryptic intron splice sites alone had no impact on siRNA production from *cenH* and H3K9me, due to Swi6^{HP1} independently recruiting RDRP. However, when *crypticSS^{mut}* was combined with *swi6Δ*, siRNAs and H3K9me were abolished (Fig. 6b, d). This remarkable observation mimics the effect of combining *swi6Δ* with deletions of TFs or a mutation in the spliceosome subunit Cwf10. Together, these findings indicate that TF-driven bottom-strand transcripts engage spliceosome machinery through cryptic introns to generate siRNAs that are required for initial heterochromatin formation. We have added further clarification in the Discussion section (page 17, first paragraph).

4. *To enhance clarity and impact, the authors could consider moving Supplementary Figure 4f closer to Figure 4g. This would emphasize that only the bottom transcript is affected and help visually connect this key finding.*

We thank the reviewer for the helpful suggestion. The top strands are now shown alongside the bottom strands (Fig. 5c, d).

Response to Reviewer Comments:

We appreciate the valuable feedback of the reviewers and we thank you for your help with the review process.

Please find below our detailed response to referees' comments:

Reviewer #1 (Remarks to the Author):

The authors have addressed all of my concerns and the manuscript appears ready for publication.

We thank the reviewer for insightful comments during the revision process.

Reviewer #2 (Remarks to the Author):

With the new results, the work is robust. Additionally, the inclusion of replicates strengthens the conclusions. I believe the paper will be published.

Minor comment:

For Fig. 4e–h and Supplementary Fig. 5, I suggest adding a quantification of the effect. This is particularly important for the mutant of Php5, as the mutation might completely abolish the binding of the trimer to DNA.

We are glad that the reviewer was convinced of the reproducibility of our findings, and we appreciate this suggestion. We have stated a quantification of the effect in the text as follows: "Indeed, site-specific ChIP-qPCR analysis revealed a ~90% reduction in Php3 and a ~80% reduction in Php5 in HFD mutants compared to WT".

Again, we thank the reviewer for the constructive criticism provided during the revision process.

Reviewer #4 (Remarks to the Author):

The authors have successfully addressed all my concerns. The paper is suitable for publication in its current format.

We thank the reviewer for valuable comments during the revision process.